# Differentiating Through Integer Linear Programs with Quadratic Regularization and Davis-Yin Splitting

**Daniel McKenzie**                                        *dmckenzie@mines.edu*
*Department of Applied Mathematics and Statistics*
*Colorado School of Mines*

**Samy Wu Fung**                                           *swufung@mines.edu*
*Department of Applied Mathematics and Statistics*
*Department of Computer Science*
*Colorado School of Mines*

**Howard Heaton**                                          *research@typal.academy*
*Typal Academy*

**Reviewed on OpenReview:** *https://openreview.net/forum?id=H8IaxrANWl&noteId*

## Abstract

In many applications, a combinatorial problem must be repeatedly solved with similar, but distinct parameters. Yet, the parameters $w$ are not directly observed; only contextual data $d$ that correlates with $w$ is available. It is tempting to use a neural network to predict $w$ given $d$. However, training such a model requires reconciling the discrete nature of combinatorial optimization with the gradient-based frameworks used to train neural networks. We study the case where the problem in question is an Integer Linear Program (ILP). We propose applying a three-operator splitting technique, also known as Davis-Yin splitting (DYS), to the quadratically regularized continuous relaxation of the ILP. We prove that the resulting scheme is compatible with the recently introduced Jacobian-free back-propagation (JFB). Our experiments on two representative ILPs: the shortest path problem and the knapsack problem, demonstrate that this combination—DYS on the forward pass, JFB on the backward pass—yields a scheme which scales more effectively to high-dimensional problems than existing schemes. All code associated with this paper is available at `https://github.com/mines-opt-ml/fpo-dys`.

## 1 Introduction

Many high-stakes decision problems in healthcare (Zhong & Tang, 2021), logistics and scheduling (Sbihi & Eglese, 2010; Kacem et al., 2021), and transportation (Wang & Tang, 2021) can be viewed as a two step process. In the first step, one gathers data about the situation at hand. This data is used to assign a value (or cost) to outcomes arising from each possible action. The second step is to select the action yielding maximum value (alternatively, lowest cost). Mathematically, this can be framed as an optimization problem with a data-dependent cost function:

$$x^\star(d) \triangleq \underset{x \in \mathcal{X}}{\arg\min} \, f(x; d). \tag{1}$$

In this work, we focus on the case where $\mathcal{X} \subset \mathbb{R}^n$ is a finite constraint set and $f(x; d) = w(d)^\top x$ is a linear function. This class of problems is quite rich, containing the shortest path, traveling salesperson, and sequence alignment problems, to name a few. Given $w(d)$, solving equation 1 may be straightforward (*e.g.* shortest path) or NP-hard (*e.g.* traveling salesperson problem (Karp, 1972)). However, our present interest is settings where the dependence of $w(d)$ on $d$ is *unknown*. In such settings, it is intuitive to *learn a mapping*

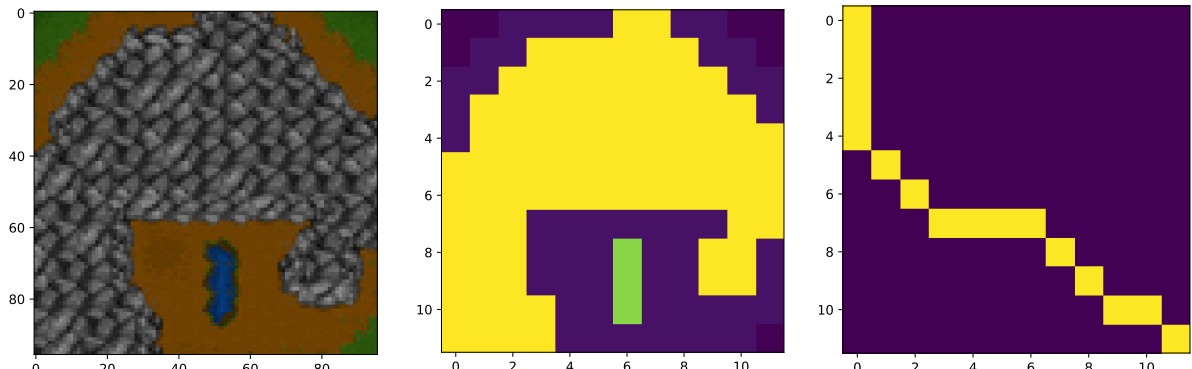

Figure 1: The shortest path prediction problem (Pogančić et al., 2019). The goal is to find the shortest path (from top-left to bottom-right) through a randomly generated terrain map from the Warcraft II tileset (Guyomarch). The contextual data $d$, shown in (a), is an image sub-divided into 8-by-8 squares, each representing a vertex in a 12-by-12 grid graph. The cost of traversing each square, *i.e.* $w(d)$, is shown in (b), with darker shading representing lower cost. The true shortest path is shown in (c).

$w_\Theta(d) \approx w(d)$ and then solve

$$x_\Theta(d) \triangleq \underset{x \in \mathcal{X}}{\arg\min}\, w_\Theta(d)^\top x \tag{2}$$

in lieu of $w(d)$. The observed data $d$ is called the *context*. As an illustrative running example, consider the shortest path prediction problem shown in Figure 1, which is studied in Berthet et al. (2020) and Pogančić et al. (2019).

At first glance, it may appear natural to tune the weights $\Theta$ so as to minimize the difference between $w_\Theta(d)$ and $w(d)$. However, this is only possible if $w(d)$ is available at training time. Even if this approach is feasible, it may not be advisable. If $w_\Theta(d)$ is a near-perfect predictor of $w(d)$ we can expect $x_\Theta(d) \approx x^\star(d)$. However, if there are even small differences between $w_\Theta(d)$ and $w(d)$ this can manifest in wildly different solutions (Bengio, 1997; Wilder et al., 2019). Thus, we focus on methods where $\Theta$ is tuned by minimizing the discrepancy between $x_\Theta(d)$ and $x^\star(d)$. Such methods are instances of the decision-focused learning paradigm Mandi et al. (2023), as the criterion we are optimizing for is the quality of $x_\Theta(d)$ (the "decision") not the fidelity of $w_\Theta(d)$ (the "prediction").

The key obstacle in using gradient-based algorithms to tune $\Theta$ in such approaches is "differentiating through" the solution $x_\Theta(d)$ of equation 2 to obtain a gradient with which to update $\Theta$. Specifically, the combinatorial nature of $\mathcal{X}$ may cause the solution $x_\Theta(d)$ to remain unchanged for many small perturbations to $\Theta$; yet, for some perturbations $x_\Theta(d)$ may "jump" to a different point in $\mathcal{X}$. Hence, the gradient $\mathrm{d}x_\Theta/\mathrm{d}w_\Theta$ is always either zero or undefined. To compute an informative gradient, we follow Wilder et al. (2019) by relaxing equation 2 to a quadratic program over the convex hull of $\mathcal{X}$ with an added regularizer (see equation 8).

**Contribution** Drawing upon recent advances in convex optimization (Ryu & Yin, 2022) and implicit neural networks (Fung et al., 2022; McKenzie et al., 2021), we propose a method designed specifically for "differentiating through" large-scale integer linear programs. Our approach is fast, easy to implement using our provided code, and, unlike several prior works (*e.g.* see Pogančić et al. (2019); Berthet et al. (2020)), trains completely on GPU. Numerical examples herein demonstrate our approach, run using only standard computing resources, easily scales to problems with tens of thousands of variables. Theoretically, we verify our approach computes an informative gradient via a refined analysis of Jacobian-free Backpropagation (JFB) (Fung et al., 2022). In summary, we do the following.

    ▷ Building upon McKenzie et al. (2021), we use the three operator splitting technique of Davis & Yin (2017) to propose `DYS-Net`.

▷ We provide theoretical guarantees for differentiating through the fixed point of a certain non-expansive, but not contractive, operator.

▷ We numerically show `DYS-Net` easily handles combinatorial problems with tens of thousands of variables.

## 2 Preliminaries

**LP Reformulation**  We focus on optimization problems of the form equation 1 where $f(x; d) = w(d)^\top x$ and $\mathcal{X}$ is the integer or binary points of a polytope, which we may assume to be expressed in standard form (Ziegler, 2012):

$$\mathcal{X} = \mathcal{C} \cap \mathbb{Z}^n \quad \text{or} \quad \mathcal{X} = \mathcal{C} \cap \{0, 1\}^n, \quad \text{where} \quad \mathcal{C} = \{x \in \mathbb{R}^n : Ax = b \text{ and } x \geq 0\}. \tag{3}$$

In other words, equation 1 is an Integer Linear Program (ILP). Similar to works by Elmachtoub & Grigas (2022); Wilder et al. (2019); Mandi & Guns (2020) and others we replace the model equation 2 with its continuous relaxation, and redefine

$$x_\Theta(d) \triangleq \underset{x \in \mathcal{C}}{\arg\min} \, w_\Theta(d)^\top x \tag{4}$$

as a step towards making the computation of $\mathrm{d}x_\Theta / \mathrm{d}w_\Theta$ feasible.

**Losses and Training Data**  We assume access to training data in the tuple form $(d, x^\star(d))$. Such data may be obtained by observing experienced agents solve equation 1, for example taxi drivers in a shortest path problem Piorkowski et al. (2009). We focus on the $\ell_2$ loss:

$$\mathcal{L}(\Theta) \triangleq \mathbb{E}_{d \sim \mathcal{D}} \left[ \ell_2(\Theta; d) \right], \quad \text{where} \quad \ell_2(\Theta; d) = \|x^\star(d) - x_\Theta(d)\|^2, \tag{5}$$

and $\mathcal{D}$ is the distribution of contextual data, although more sophisticated losses can be used with only superficial changes to our results. In principle we select the optimal weights by solving $\arg\min_\Theta \mathcal{L}(\Theta)$. In practice, the population risk is inaccessible, and so we minimize empirical risk instead (Vapnik, 1999):

$$\underset{\Theta}{\arg\min} \, \frac{1}{N} \sum_{i=1}^{N} \ell_2\left(\Theta; d_i\right). \tag{6}$$

**Argmin Differentiation**  For notational brevity, we temporarily suppress the dependence on $d$. The gradient of $\ell_2$ with respect to $\Theta$, evaluated on a single data point is

$$\frac{\mathrm{d}}{\mathrm{d}\Theta} \left[ \ell_2(\Theta) \right] = \frac{\mathrm{d}}{\mathrm{d}\Theta} \left[ \|x_\Theta - x^\star\|^2 \right] = (x_\Theta - x^\star)^\top \frac{\partial x_\Theta}{\partial w_\Theta} \frac{\mathrm{d}w_\Theta}{\mathrm{d}\Theta}.$$

As discussed in Section 1, $x_\Theta^\star$ is piecewise constant as a function of $w_\Theta$, and this remains true for the LP relaxation equation 4. Consequently, for all $w_\Theta$, either $\partial x_\Theta / \partial \Theta = 0$ or $\partial x_\Theta / \partial \Theta$ is undefined—neither case yields an informative gradient. To remedy this, Wilder et al. (2019); Mandi & Guns (2020) propose adding a small amount of regularization to the objective function in equation 4 to make it strongly convex. This ensures $x_\Theta$ is a continuously differentiable function of $w_\Theta$. We add a small quadratic regularizer, modulated by $\gamma \geq 0$, to obtain the objective

$$f_\Theta(x; \gamma, d) \triangleq w_\Theta(d)^\top x + \gamma \|x\|_2^2 \tag{7}$$

and henceforth replace equation 4 by

$$x_\Theta(d) \triangleq \underset{x \in \mathcal{C}}{\arg\min} \, f_\Theta(x; \gamma, d). \tag{8}$$

During training, we aim to solve equation 8 *and* efficiently compute the derivative $\partial x_\Theta / \partial \Theta$.

**Pitfalls of Relaxation**  Finally, we offer a note of caution when using the quadratically regularized problem equation 8 as a proxy for the original integer linear program equation 1. Temporarily defining

$$\hat{x}_\Theta(d) \triangleq \arg\min_{x \in \mathcal{X}} w_\Theta(d)^\top x \quad \tilde{x}_\Theta(d) \triangleq \arg\min_{x \in \mathcal{C}} w_\Theta(d)^\top x \quad x_\Theta(d) \triangleq \arg\min_{x \in \mathcal{C}} f_\Theta(x; \gamma, d),$$

we observe that two sources of error have arisen; one from replacing $\mathcal{X}$ with $\mathcal{C}$ (i.e. the discrepancy between $\hat{x}_\Theta$ and $\tilde{x}_\Theta$), and one from replacing the linear objective function with its quadratically regularized counterpart (*i.e.* the discrepancy between $\tilde{x}_\Theta$ and $x_\Theta$). When $A$ is *totally unimodular*, $\tilde{x}_\Theta$ is guaranteed to be integral, in which case $\hat{x}_\Theta = \tilde{x}_\Theta$. Moreover, Wilder et al. (2019, Theorem 2) shows that

$$0 \leq w_\Theta(d)^\top \left(x_\Theta(d) - \tilde{x}_\Theta(d)\right) \leq \gamma \max_{x,y \in \mathcal{C}} \|x - y\|^2$$

whence, as in the proof Berthet et al. (2020, Proposition 2.2), $\lim_{\gamma \to 0^+} x_\Theta(d) = \tilde{x}_\Theta(d)$ (at least when $\mathcal{C}$ is compact). Thus, for totally unimodular ILPs (*e.g.* the shortest path problem), it is reasonable to expect that $x_\Theta(d)$ obtained from equation 8, for well-tuned $\Theta$ and small $\gamma$, are of acceptable quality. For ILPs which are not totally unimodular no such theoretical guarantees exist. However, we observe that this relaxation works reasonably well, at least for the knapsack problem.

## 3    Related Works

We highlight recent works in decision-focused learning—particularly for mixed integer programs—and related areas.

**Optimization Layers**  Introduced in Amos & Kolter (2017), and studied further in Agrawal et al. (2019b;c); Bolte et al. (2021); Blondel et al. (2022), an *optimization layer* is a modular component of a deep learning framework where forward propagation consists of solving a parametrized optimization problem. Consequently, backpropagation entails differentiating the solution to this problem with respect to the parameters. Optimization layers are a promising technology as they are able to incorporate hard constraints into deep neural networks. Moreover, as their output is a (constrained) minimizer of the objective function, it is easier to interpret than the output of a generic layer.

**Decision-focused learning for LPs**  When an optimization layer is fitted to a data-dependent optimization problems of the form equation 1, with the goal of maximizing the quality of the predicted solution, this is usually referred to as *decision-focused learning.* We narrow our focus to the case where the objective function in equation 1 is linear.Wilder et al. (2019) and Elmachtoub & Grigas (2022) are two of the earliest works applying deep learning techniques to data-driven LPs, and delineate two fundamentally different approaches. Specifically, Wilder et al. (2019) proposes replacing the LP with a continuous and strongly convex relaxation, as described in Section 2. This approach is extended to ILPs in Ferber et al. (2020), and to non-quadratic regularizers in Mandi & Guns (2020). On the other hand, Elmachtoub & Grigas (2022) avoid modifying the underlying optimization problem and instead propose a new loss function; dubbed the Smart Predict-then-Optimize (SPO) loss, to be used instead of the $\ell_2$ loss. We emphasize that the SPO loss requires access to the true cost vectors $w(d)$, a setting which we do not consider.

Several works define a continuously differentiable proxy for the solution to the *unregularized* LP equation 4, which we rewrite here as[1]

$$x^\star(w) = \arg\min_{x \in \mathcal{C}} w^\top x, \tag{9}$$

In Berthet et al. (2020), a stochastic perturbation is considered:

$$x^\star_\varepsilon(w) = \mathbb{E}_Z \left[\arg\min_{x \in \mathcal{C}} (w + \varepsilon Z)^\top x\right], \tag{10}$$

which is somewhat analogous to Nesterov-Spokoiny smoothing (Nesterov & Spokoiny, 2017) in zeroth-order optimization. For some draws of $Z$ the additively perturbed cost vector $w + \varepsilon z$ may have negative entries,

---

[1]For sake of notational simplicity, the dependence on $d$ is implicit.

which may cause a solver (*e.g.* Dijkstra's algorithm) applied to the associated LP to fail. To avoid this, Dalle et al. (2022) suggest using a *multiplicative perturbation* of $w$ instead. Pogančić et al. (2019) define a piecewise-affine interpolant to $\ell(x^\star(w))$, where $\ell$ is a suitable loss function.

We note a bifurcation in the literature; Wilder et al. (2019); Ferber et al. (2020); Elmachtoub & Grigas (2022) assume access to training data of the form $(d, w(d))$, whereas Pogančić et al. (2019); Berthet et al. (2020); Sahoo et al. (2022) use training data of the form $(d, x^\star(d))$. The former setting is frequently referred to as the *predict-then-optimize* problem. Our focus is on the latter setting. We refer the reader to Kotary et al. (2021); Mandi et al. (2023) for recent surveys of this area.

**Deep Equilibrium Models**  Bai et al. (2019); El Ghaoui et al. (2021) propose the notion of *deep equilibrium model* (DEQ), also known as an *implicit neural network*. A DEQ is a neural network for which forward propagation consists of (approximately) computing a fixed point of a parametrized operator. We note that equation 8 may be reformulated as a fixed point problem,

$$\text{Find } x_\Theta \text{ such that } x_\Theta = P_\mathcal{C}\left(x_\Theta - \alpha \nabla_x f_\Theta(x_\Theta; d)\right), \tag{11}$$

where $P_\mathcal{C}$ is the orthogonal projection[2] onto $\mathcal{C}$. Thus, DEQ techniques may be applied to constrained optimization layers (Chen et al., 2021; Blondel et al., 2022; McKenzie et al., 2021). However, the cost of computing $P_\mathcal{C}$ can be prohibitive, see the discussion in Section 4.

**Learning-to-Optimize (L2O)**  Our work is related to the learning-to-optimize (L2O) framework (Chen et al., 2022; Li & Malik, 2017; Chen et al., 2018; Liu et al., 2023), where an optimization algorithm is learned and its outputs are used to perform inferences. However, traditional L2O methods use a fixed number of layers (*i.e.* unrolled algorithms). Our approach attempts to differentiate through the solution $x_\Theta^\star$ and is therefore most closely aligned with works that employ implicit networks within the L2O framework (Amos & Kolter, 2017; Heaton & Fung, 2023; McKenzie et al., 2021; Gilton et al., 2021; Liu et al., 2022). We highlight that, unlike many L2O works, our goal is not to learn a faster optimization method for fully specified problems. Rather, we seek to solve partially specified problems given contextual data by combining learning and optimization techniques.

**Computing the derivative of a minimizer with respect to parameters**  In all of the aforementioned works, the same fundamental problem is encountered: $\partial x_\Theta / \partial \Theta$ must be computed where $x_\Theta$ is the solution of a (constrained) optimization problem with objective function parametrized by $\Theta$. The most common approach to doing so, proposed in Amos & Kolter (2017) and used in Ruthotto et al. (2018); Wilder et al. (2019); Mandi & Guns (2020); Ferber et al. (2020), starts with the KKT conditions for constrained optimality:

$$\frac{\partial f_\Theta}{\partial x}(x_\Theta) + A^\top \hat{\lambda} + \hat{\nu} = 0, \ Ax - b = 0, \ D(\hat{\nu})x_\Theta = 0,$$

where $\hat{\lambda}$ and $\hat{\nu} \geq 0$ are Lagrange multipliers associated to the optimal solution $x_\Theta$ (Bertsekas, 1997) and $D(\hat{\nu})$ is a matrix with $\hat{\nu}$ along its diagonal. Differentiating these equations with respect to $\Theta$ and rearranging yields

$$\begin{bmatrix} \frac{\partial^2 f_\Theta}{\partial x^2} & A & I \\ A^\top & 0 & 0 \\ D(\hat{\nu}) & 0 & D(x_\Theta) \end{bmatrix} \begin{bmatrix} \frac{dx_\Theta}{d\Theta} \\ \frac{d\hat{\lambda}}{d\Theta} \\ \frac{d\hat{\nu}}{d\Theta} \end{bmatrix} = \begin{bmatrix} \frac{\partial^2 f_\Theta}{\partial x \partial \Theta} \\ 0 \\ 0 \end{bmatrix}. \tag{12}$$

The matrix and right hand side vector in equation 12 are computable, thus enabling one to solve for $\frac{dx_\Theta}{d\Theta}$ (as well as $\frac{d\hat{\lambda}}{d\Theta}$ and $\frac{d\hat{\nu}}{d\Theta}$). We emphasize that both primal (*i.e.*, $x_\Theta$) and dual (*i.e.*, $\hat{\lambda}$ and $\hat{\nu}$) variables at optimality are required. This constrains the choice of optimization schemes to those that track primal and dual variables, and prohibits the use of fast, primal-only schemes. Following (Amos & Kolter, 2017), most works employing this approach use a primal-dual interior point method, which computes acceptable approximations to $x_\Theta$, $\hat{\lambda}$, and $\hat{\nu}$ at a cost of $\mathcal{O}\left(\max\{n, m\}^3\right)$, assuming $x \in \mathbb{R}^n$ and $A \in \mathbb{R}^{m \times n}$.

Using the stochastic proxy equation 10, Berthet et al. (2020) derive a formula for $dx_\varepsilon^\star / dw$ which is also an

---

[2]For a set $\mathcal{A} \subseteq \mathbb{R}^n$, $P_\mathcal{A}(x) \triangleq \arg\min_{z \in \mathcal{A}} \|z - x\|$.

expectation, and hence can be efficiently approximated using Monte Carlo methods. Pogančić et al. (2019) show that the gradients of their interpolant are strikingly easy to compute, requiring just one additional solve of equation 9 with perturbed cost $w'$. This approach is extended by Sahoo et al. (2022) which proposes to avoid computing $dx_\varepsilon^\star/dw$ entirely, replacing it with the negative identity matrix. This is similar in spirit to our use of Jacobian-free Backpropagation (see Section 4).

## 4 DYS-Net

We now introduce our proposed model, DYS-net. We use this term to refer to the model *and* the training procedure. Fixing an architecture for $w_\Theta$, and an input $d$, DYS-net computes an approximation to $x_\Theta(d)$ in a way that is easy to backpropagate through:

$$\texttt{DYS-net}(d;\ \Theta) \approx x_\Theta \triangleq \arg\min_{x \in \mathcal{C}} f_\Theta(x; \gamma, d). \tag{13}$$

**The Forward Pass**  As we wish to compute $x_\Theta$ and $\partial x_\Theta/\partial\Theta$ for high dimensional settings (*i.e.* large $n$), we eschew second-order methods (*e.g.* Newton's method) in favor of first-order methods such as projected gradient descent (PGD). With PGD, a sequence $\{x^k\}$ of estimates of $x_\Theta$ are computed so that $x_\Theta = \lim_{k\to\infty} x^k$ where

$$x^{k+1} = P_\mathcal{C}\left(x^k - \alpha\nabla_x f(x^k; \gamma, d)\right) \quad \text{for all } k \in \mathbb{N}.$$

This approach works for simple sets $\mathcal{C}$ for which there exists an explicit form of $P_\mathcal{C}$, *e.g.* when $\mathcal{C}$ is the probability simplex (Duchi et al., 2008; Condat, 2016; Li et al., 2023). However, for general polytopes $\mathcal{C}$ no such form exists, thereby requiring a second iterative procedure to compute $P_\mathcal{C}(x^k)$. This projection must be computed at every iteration of every forward pass for every sample of every epoch, and this dominates the computational cost, see McKenzie et al. (2021, Appendix 4).

To avoid this expense, we draw on recent advances in the convex optimization literature, particularly Davis-Yin splitting or DYS (Davis & Yin, 2017). DYS is a three operator splitting technique, and hence is able to decouple the polytope constraints into relatively simple constraints Davis & Yin (2017). This technique has been used elsewhere, *e.g.* Pedregosa & Gidel (2018); Yurtsever et al. (2021). Specifically, we adapt the architecture incorporating DYS proposed in McKenzie et al. (2021). To this end, we rewrite $\mathcal{C}$ as

$$\mathcal{C} = \{x :\ Ax = b \text{ and } x \geq 0\} = \underbrace{\{x :\ Ax = b\}}_{\triangleq\, \mathcal{C}_1} \cap \underbrace{\{x :\ x \geq 0\}}_{\triangleq\, \mathcal{C}_2} = \mathcal{C}_1 \cap \mathcal{C}_2. \tag{14}$$

While $P_\mathcal{C}$ is hard to compute, both $P_{\mathcal{C}_1}$ and $P_{\mathcal{C}_2}$ can be computed cheaply via explicit formulas, once a singular value decomposition (SVD) is computed for $A$. We verify this via the following lemma (included for completeness, as the two results are already known).

**Lemma 1.** *If $\mathcal{C}_1, \mathcal{C}_2$ are as in equation 14 and $A$ is full-rank then:*

1. $P_{\mathcal{C}_1}(z) = z - A^\dagger(Az - b)$, *where $A^\dagger = V\Sigma^{-1}U^\top$ and $U\Sigma V^\top$ is the compact SVD of $A$.*

2. $P_{\mathcal{C}_2}(z) = \mathrm{ReLU}(z) \triangleq \max\{0, z\}$.

We remark that the SVD of $A$ is computed offline, and is a once-off cost. Alternatively, one could compute $P_{\mathcal{C}_1}$ by using an iterative method (*e.g.* GMRES) to find the (least squares) solution to $Av = Az - b$, whence $P_{\mathcal{C}_1}(z) = z - v$, but this would need to be done at each iteration of each forward pass for every sample of every epoch. Thus, it is unclear whether this procides computational savings over the once-off computation of the SVD. However, this could prove useful when $A$ is only provided implicitly, *i.e.* via access to a matrix-vector product oracle.

The following theorem allows one to approximate $x_\Theta$ using only $P_{\mathcal{C}_1}$ and $P_{\mathcal{C}_2}$, not $P_\mathcal{C}$.

**Theorem 2.** *Let $\mathcal{C}_1, \mathcal{C}_2$ be as in equation 14, and suppose $f_\Theta(x; \gamma, d) = w_\Theta(d)^\top x + \frac{\gamma}{2}\|x\|_2^2$ for any neural network $w_\Theta(d)$. For any $\alpha \in (0, 2/\gamma)$ define the sequence $\{z^k\}$ by:*

$$z^{k+1} = T_\Theta(z^k) \quad \text{for all } k \in \mathbb{N} \tag{15}$$

*where*

$$T_\Theta(z) \triangleq z - P_{\mathcal{C}_2}(z) + P_{\mathcal{C}_1}\left((2 - \alpha\gamma)\, P_{\mathcal{C}_2}(z) - z - \alpha w_\Theta(d)\right). \tag{16}$$

*If $x^k \triangleq P_{\mathcal{C}_2}(z^k)$ then the sequence $\{x^k\}$ converges to $x_\Theta$ in equation 8 and $\|x^{k+1} - x^k\|_2^2 = O(1/k)$.*

A full proof is included in Appendix A; here we provide a proof sketch. Substituting the gradient

$$\nabla_z f_\Theta(z; \gamma, d) = w_\Theta(d) + \gamma z$$

for $F_\Theta(\cdot, d)$ into the formula for $T_\Theta(\cdot)$ given in McKenzie et al. (2021, Theorem 3.2) and rearranging yields equation 16. Verifying the assumptions of McKenzie et al. (2021, Theorem 3.3) are satisfied is straightforward. For example, $\nabla_z f_\Theta(z; \gamma, d)$ is $1/\gamma$-cocoercive by the Baillon-Haddad theorem (Baillon & Haddad, 1977; Bauschke & Combettes, 2009).

The forward pass of `DYS-net` consists of iterating equation 15 until a suitable stopping criterion is met. For simplicity, in presenting Algorithm 1 we assume this to be a maximum number of iterations is reached. One may also consider a stopping criterion of $\|z_{k+1} - z_k\| \leq$ tol.

**The Backward Pass** From Theorem 2, we deduce the following fixed point condition:

$$x_\Theta = P_{\mathcal{C}_2}(z_\Theta), \quad \text{for some } z_\Theta \in \{z : z = T_\Theta(z)\}. \tag{17}$$

As discussed in McKenzie et al. (2021), instead of backpropagating through every iterate of the forward pass, we may derive a formula for $dx_\Theta/d\Theta$ by appealing to the implicit function theorem and differentiating both sides of equation 17:

$$\frac{\mathrm{d}z_\Theta}{\mathrm{d}\Theta} = \frac{\partial T_\Theta}{\partial \Theta} + \frac{\partial T_\Theta}{\partial z}\frac{\mathrm{d}z_\Theta}{\mathrm{d}\Theta} \quad \implies \quad \mathcal{J}_\Theta(z_\Theta)\,\frac{\mathrm{d}z_\Theta}{\mathrm{d}\Theta} = \frac{\partial T_\Theta}{\partial \Theta} \text{ where } \mathcal{J}_\Theta(z) = I - \frac{\partial T_\Theta}{\partial z}. \tag{18}$$

We notice two immediate problems not addressed by McKenzie et al. (2021): (i) $T_\Theta$ is not everywhere differentiable with respect to $z$, as $P_{\mathcal{C}_2}$ is not; (ii) if $T_\Theta$ were a contraction (*i.e.* Lipschitz in $z$ with constant less than 1), then $\mathcal{J}_\Theta$ would be invertible. However, this is not necessarily the case. Thus, it is not clear *a priori* that equation 18 can be solved for $\mathrm{d}z_\Theta/\mathrm{d}\Theta$. Our key result (Theorem 6 below) is to provide reasonable conditions under which $\mathcal{J}_\Theta(z_\Theta)$ is invertible.

Assuming these issues can be resolved, one may compute the gradient of the loss using the chain rule:

$$\frac{\mathrm{d}\ell}{\mathrm{d}\Theta} = \frac{\mathrm{d}\ell}{\mathrm{d}x}\frac{\mathrm{d}x_\Theta}{\mathrm{d}\Theta} = \frac{\mathrm{d}\ell}{\mathrm{d}x}\left(\frac{\mathrm{d}P_{\mathcal{C}_2}}{\mathrm{d}z}\frac{\mathrm{d}z_\Theta}{\mathrm{d}\Theta}\right) = \frac{\mathrm{d}\ell}{\mathrm{d}x}\frac{\mathrm{d}P_{\mathcal{C}_2}}{\mathrm{d}z}\mathcal{J}_\Theta^{-1}\frac{\partial T_\Theta}{\partial \Theta}$$

This approach requires solving a linear system with $\mathcal{J}_\Theta$ which becomes particularly expensive when $n$ is large. Instead, we use *Jacobian-free Backpropagation* (JFB) (Fung et al., 2022), also known as one-step differentiation (Bolte et al., 2024), in which the Jacobian $\mathcal{J}_\Theta$ is replaced with the identity matrix. This leads to an approximation of the true gradient $\mathrm{d}\ell/\mathrm{d}\Theta$ using

$$p_\Theta(x_\Theta) = \left[\frac{\partial \ell}{\partial x}\frac{\mathrm{d}P_{\mathcal{C}_2}}{\mathrm{d}z}\frac{\partial T_\Theta}{\partial \Theta}\right]_{(x,z)=(x_\Theta, z_\Theta)}. \tag{19}$$

This update can be seen as a zeroth order Neumann expansion of $\mathcal{J}_\Theta^{-1}$ Fung et al. (2022); Bolte et al. (2024). We show equation 19 is a valid descent direction by resolving the two problems highlighted above. We begin by rigorously deriving a formula for $\partial T_\Theta/\partial z$. Recall the following generalization of the Jacobian to non-smooth operators due to Clarke (1983).

---

**Algorithm 1** `DYS-Net`

---

1: **Inputs:** $A$ and $b$ defining $\mathcal{C}$, hyperparameters $\alpha, \gamma, K$
2: **Initialize:** Attach neural network $w_\Theta(\cdot)$. Compute SVD of $A$ for $P_{\mathcal{C}_2}$ formula
3: **Forward Pass:** Initialize $z^0$. Given $w_\Theta(d)$ compute $z^1, \ldots, z^K$ using equation 15. Return $x^K \triangleq P_{\mathcal{C}_1}(z^K) \approx x_\Theta$.
4: **Backward Pass:** Compute $p_\Theta(x^K) \approx p_\Theta(x_\Theta)$ using equation 19 and return.

---

**Definition 3.** *For any locally Lipschitz $F : \mathbb{R}^n \to \mathbb{R}^n$ let $D_F$ denote the set upon which $F$ is differentiable. The Clarke Jacobian of $F$ is the set-valued function defined as*

$$\partial F(\bar{z}) = \left\{ \begin{array}{ll} \left.\frac{\mathrm{d}F}{\mathrm{d}z}\right|_{z=\bar{z}} & \text{if } \bar{z} \in D_F \\ \text{Con}\left\{\lim_{z' \to \bar{z}: z' \in D_F} \left.\frac{\mathrm{d}F}{\mathrm{d}z}\right|_{z=z'}\right\} & \text{if } \bar{z} \notin D_F \end{array} \right.$$

*Where* $\text{Con}\{\}$ *denotes the convex hull of a set.*

The Clarke Jacobian of $P_{\mathcal{C}_2}$ is easily computable, see Lemma 9. Define the (set-valued) functions

$$c(\alpha) \triangleq \partial \max(0, \alpha) = \left\{ \begin{array}{ll} 1 & \text{if } \alpha > 0 \\ 0 & \text{if } \alpha < 0 \\ [0,1] & \text{if } \alpha = 0 \end{array} \right. \quad \text{and} \quad \tilde{c}(\alpha) = \left\{ \begin{array}{ll} 1 & \text{if } \alpha > 0 \\ 0 & \text{if } \alpha \le 0 \end{array} \right.$$

Then

$$\partial P_{\mathcal{C}_2}(\bar{z}) = \left[\frac{\mathrm{d}}{\mathrm{d}z} \text{ReLU}(z)\right]_{z=\bar{z}} = \text{diag}(c(\bar{z})), \tag{20}$$

where $c$ is applied element-wise. We shall now provide a consistent rule for selecting, at any $\bar{z}$, an element of $\partial P_{\mathcal{C}_2}(\bar{z})$. If $z_i \ne 0$ for all $i$ then $\partial P_{\mathcal{C}_2}$ is a singleton. If one or more $z_i = 0$ then $\partial P_{\mathcal{C}_2}$ is multi-valued, so we choose the element of $\partial P_{\mathcal{C}_2}$ with 0 in the $(i,i)$ position for every $z_i = 0$. We write $dP_{\mathcal{C}_2}/dz$ for this chosen element, and note that

$$\left.\frac{\mathrm{d}P_{\mathcal{C}_2}}{\mathrm{d}z}\right|_{z=\bar{z}} = \text{diag}(\tilde{c}(\bar{z})) \in \partial P_{\mathcal{C}_2}(\bar{z})$$

This aligns with the default rule for assigning a sub-gradient to ReLU used in the popular machine learning libraries TensorFlow(Abadi et al., 2016), PyTorch (Paszke et al., 2019) and JAX (Bradbury et al., 2018), and has been observed to yield networks which are more stable to train than other choices (Bertoin et al., 2021).

Given the above convention, we can compute $\partial T_\Theta / \partial z$, interpreted as an element of the Clarke Jacobian of $T_\Theta$ chosen according to a consistent rule. Surprisingly, $\partial T_\Theta / \partial z$ may be expressed using only orthogonal projections to hyperplanes. Throughout, we let $e_i \in \mathbb{R}^n$ be the one-hot vector with 1 in the $i$-th position and zeros elsewhere, and $a_i^\top$ be the $i$-th row of $A$. For any subspace $\mathcal{H}$ we denote the orthogonal subspace as $\mathcal{H}^\perp$. The following theorem, the proof of which is given in Appendix A, characterizes $\partial T_\Theta / \partial z$.

**Theorem 4.** *Suppose $A$ is full-rank and $\mathcal{H}_1 \triangleq \text{Null}(A)$, $\mathcal{H}_{2,z} \triangleq \text{Span}(e_i : z_i > 0)$. Then for all $\hat{z} \in \mathbb{R}^n$*

$$\left.\frac{\partial T_\Theta}{\partial z}\right|_{z=\bar{z}} = P_{\mathcal{H}_1^\perp} P_{\mathcal{H}_{2,\bar{z}}^\perp} + (1 - \alpha\gamma) \cdot P_{\mathcal{H}_1} P_{\mathcal{H}_{2,\bar{z}}}. \tag{21}$$

To show JFB is applicable, it suffices to verify $\|\partial T_\Theta / \partial z\| < 1$ when evaluated at $z_\Theta$. Theorem 4 enables us to show this inequality holds when $x_\Theta$ satisfies a commonly-used regularity condition, which we formalize as follows.

**Definition 5** (LICQ condition, specialized to our case)**.** *Let $x_\Theta$ denote the solution to equation 8. Let $\mathcal{A}(x_\Theta) \subseteq \{1, \ldots, n\}$ denote the set of active positivity constraints:*

$$\mathcal{A}(x_\Theta) \triangleq \{i : [x_\Theta]_i = 0\}. \tag{22}$$

*The point $x_\Theta$ satisfies the* Linear Independence Constraint Qualification *(LICQ) condition if the vectors*

$$\{a_1, \ldots, a_m\} \cup \{e_i : i \in \mathcal{A}(x_\Theta)\} \tag{23}$$

*are linearly independent.*

**Theorem 6.** *If the LICQ condition holds at $x_\Theta$, $A$ is full-rank and $\alpha \in (0, 2/\gamma)$, then $\|\partial T_\Theta / \partial z\|_{z=z_\Theta} < 1$.*

The significance of Theorem 6, which is proved in Appendix A, is outlined by the following corollary, stating that using $p_\Theta$ instead of the true gradient $d\ell/d\Theta$ is still guaranteed to decrease $\ell(\Theta)$, at least for small enough step-size.

**Corollary 7.** *If $T_\Theta$ is continuously differentiable with respect to $\Theta$ at $z_\Theta$, the assumptions in Theorem 6 hold and $(\partial T_\Theta / \partial \Theta)^\top (\partial T_\Theta / \partial \Theta)$ has condition number sufficiently small, then $p_\Theta(x_\Theta)$ is a descent direction for $\ell$ with respect to $\Theta$.*

Theorem 6 also proves that $\mathcal{J}_\Theta(z_\Theta)$ in equation 18 is invertible, thus also justifying the use of Jacobian-based backprop as well as numerous other gradient approximation techniques (Liao et al., 2018; Geng et al., 2021).

Finally, we note that in practice $p_\Theta(x^K)$ is used instead of $p_\Theta(x_\Theta)$. Fung et al. (2022, Corollary 0.1) guarantees that $p_\Theta(x^K)$ will still be a descent direction for $K$ sufficiently large, see also Bolte et al. (2024, Cor. 1).

**Synergy between forward and backward pass** We highlight that the pseudogradient computed in the backward pass (*i.e.*, $p_\Theta$) does not require any Lagrange multipliers, in contrast with the gradient computed using equation 12. This is why we may use Davis-Yin splitting, as opposed to a primal-dual interior point method, on the forward pass. This allows efficiency gains on the forward pass, as Davis-Yin splitting scales favorably with the problem dimension.

**Implementation** `DYS-net` is implemented as an abstract PyTorch (Paszke et al., 2019) layer in the provided code. To instantiate it, a user only need specify $A$ and $b$ (see equation 3) and choose an architecture for $w_\Theta$. At test time, it can be beneficial to solve the underlying combinatorial problem equation 2 exactly using a specialized algorithm, *e.g.* commercial integer programming software for the knapsack prediction problem. This can easily be done using `DYS-net` by instantiating the `test-time-forward` abstract method.

## 5 Numerical Experiments

We demonstrate the effectiveness of `DYS-Net` through four experiments. In all our experiments with `DYS-net` we used $\alpha = 0.05$ and $K = 1,000$, with an early stopping condition of $|z_{k+1} - z_k|_2 \leq$ tol with tol $= 0.01$. We compare `DYS-net` against three other methods: Perturbed Optimization (Berthet et al., 2020)—`PertOpt-net`; an approach using Blackbox Backpropagation (Vlastelica et al., 2019)—`BB-net`; and an approach using `cvxpylayers` (Agrawal et al., 2019a)—`CVX-net`. All code needed to reproduce our experiments is available at `https://github.com/mines-opt-ml/fpo-dys`.

### 5.1 Experiments using `PyEPO`

First, we verify the efficiency of `DYS-net` by performing benchmarking experiments within the `PyEPO` (Tang & Khalil, 2022) framework.

**Models** We use the `PyEPO` implementations of Perturbed Optimization, respectively Blackbox Backpropagation, as the core of `PertOpt-net`, respectively `BB-net`. We use the same architecture for $w_\Theta(d)$ for all models[3] and use an appropriate combinatorial solver at test time, via the `PyEPO` interface to `Gurobi`. Thus, the methods differ *only in training procedure*. The precise architectures used for $w_\Theta(d)$ for each problem are described in Appendix C, and are also viewable in the accompanying code repository. Interestingly,

---

[3]The architecture we use for shortest path prediction problems differs from that used for knapsack prediction problems. See the appendix for details.

we observed that adding drop-out during training to the output layer proved beneficial for the knapsack prediction problem. Without this, we find that $w_\Theta$ tends to output a sparse approximation to $w$ supported on a feasible set of items, and so does not generalize well.

**Training** We train for a maximum of 100 epochs or 30 minutes, whichever comes first. We use a validation set for model selection as we observe that, for all methods, the best loss is seldom achieved at the final iteration. Exact hyperparameters are discussed in Appendix C. For each problem size, we perform three repeats with different randomly generated data. Results are displayed in Figure 2.

**Evaluation** Given test data $\{(d_i, w(d_i), x^\star(d_i))\}_{i=1}^N$ and a fully trained model $w_\Theta(d)$ we define the *regret*:

$$r(d) = w(d_i)^\top \left( x_\Theta(d_i) - x^\star(d_i) \right). \tag{24}$$

Note that regret is non-negative, and is zero precisely when $x_\Theta(d_i)$ is a solution of equal quality to $x^\star(d_i)$. Following Tang & Khalil (2022), we evaluate the quality of $w_\Theta(d)$ using *normalized regret*, defined as

$$\tilde{r} = \frac{\sum_{i=1}^N r(d_i)}{\sum_{i=1}^N \left[ w(d_i)^\top x^\star(d_i) \right]}. \tag{25}$$

### 5.1.1 Shortest Path Prediction

The shortest path between two vertices in a graph $\mathcal{G} = (\mathcal{V}, \mathcal{E})$ can be found by:

$$x^\star = \arg\min_{x \in \mathcal{X}} w(d)^\top x; \ \mathcal{X} = \{x \in \{0,1\}^{|\mathcal{E}|} : Ex = b\}, \tag{26}$$

where $E$ is the vertex-edge adjacency matrix, $b$ encodes the initial and terminal vertices, and $w(d) \in \mathbb{R}^{|\mathcal{E}|}$ is a vector encoding ($d$-dependent) edge lengths. Here, $x^\star$ will encode the edges included in the optimal path. In this experiment we focus on the case where $\mathcal{G}$ is the $k \times k$ grid graph. We use the `PyEPO` Tang & Khalil (2022) benchmarking software to generate training datasets of the form $\{(d_i, x_i^\star \approx x^\star(d_i))\}_{i=1}^N$ where $N = 1000$ for each $k \in \{5, 10, 15, 20, 25, 30\}$. The $d_i$ are sampled from the five-dimensional multivariate Gaussian distribution with mean 0 and identity covariance. Note that the number of variables in equation 26 scales like $k^2$, not $k$.

### 5.1.2 Knapsack Prediction

In the (0–1, single) knapsack prediction problem, we are presented with a container (*i.e.* a knapsack) of size $c$ and $I$ items, of sizes $s_1, \ldots, s_I$ and values $w_1(d), \ldots, w_I(d)$. The goal is to select the subset of maximum value that fits in the container, *i.e.* to solve:

$$x^\star = \arg\max_{x \in \mathcal{X}} w(d)^\top x; \ \mathcal{X} = \{x \in \{0,1\}^I : \ s^\top x \leq c\}$$

Here, $x^\star$ encodes the selected items. In the (0–1) $k$-knapsack prediction problem we imagine the container having various notions of "size" (i.e. length, volume, weight limit) and hence a $k$-tuple of capacities $c \in \mathbb{R}^k$. Correspondingly, the items each have a $k$-tuple of sizes $s_1, \ldots, s_I \in \mathbb{R}^k$. We aim to select a subset of maximum value, amongst all subsets satisfying the $k$ capacity constraints:

$$\begin{aligned} x^\star &= \arg\max_{x \in \mathcal{X}} w(d)^\top x \\ \mathcal{X} &= \{x \in \{0,1\}^I : \ Sx \leq c\} \\ S &= \begin{bmatrix} s_1 & \cdots & s_k \end{bmatrix} \in \mathbb{R}^{k \times I} \end{aligned} \tag{27}$$

In Appendix B we discuss how to transform $\mathcal{X}$ into the canonical form discussed in Section 2. We again use `PyEPO` to generate datasets of the form $\{(d_i, x_i^\star \approx x^\star(d_i))\}_{i=1}^N$ where $N = 1000$ for $k = 2$ and $I$ varying from 50 to 750 inclusive in increments of 50. The $d_i$ from the five-dimensional multivariate Gaussian distribution with mean 0 and identity covariance.

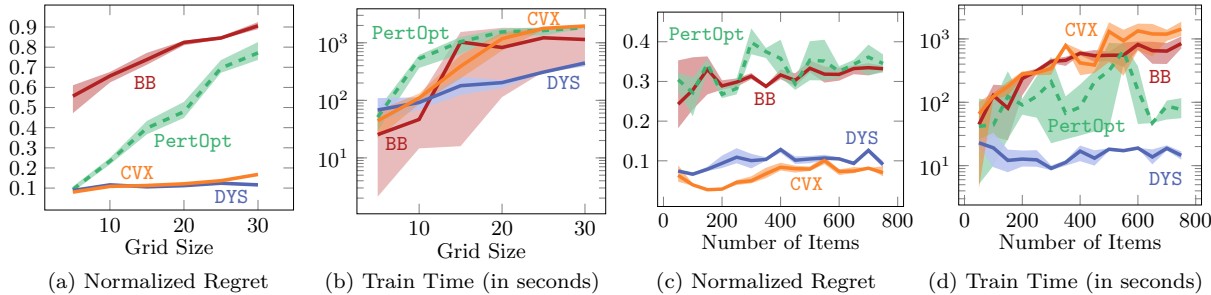

| (a) Normalized Regret | (b) Train Time (in seconds) | (c) Normalized Regret | (d) Train Time (in seconds) |

Figure 2: Results for the shortest path and knapsack prediction problems. Figures (a) and (b) show normalized regret and train time for the shortest path prediction problem, while Figures (c) and (d) show normalized regret and train time for the knapsack prediction problem. Note that the train time in figures (b) and (d) is the time till the model achieving best normalized regret on the validation set is reached.

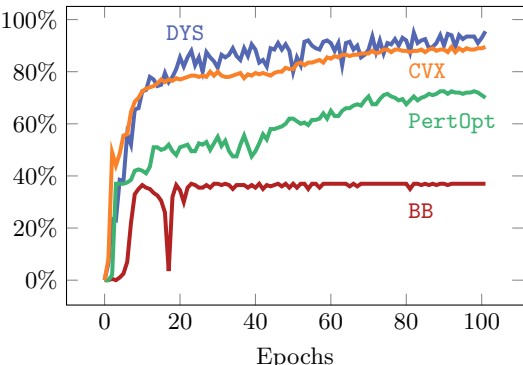

Figure 3: Accuracy (in percentage) of predicted paths on 5-by-5 grid during training.

### 5.1.3 Results

As is clear from Figure 2 `DYS-net` trains the fastest among the four methods, and achieves the best or near-best normalized regret in all cases. We attribute the accuracy to the fact that it is *not* a zeroth order method. Moreover, avoiding computation and backpropagation through the projection $P_{\mathcal{C}}$ allows DYS to have fast runtimes.

### 5.2 Large-scale shortest path prediction

Our shortest path prediction experiment described in Section 5.1.1 is bottlenecked by the fact that the "academic" license of `Gurobi` does not allow for shortest path prediction problems on grids larger than $30{\times}30$. To further explore the limits of `DYS-net`, we redo this experiment using a custom `pyTorch` implementation of Dijkstra's algorithm[4] as our base solver for equation 26.

**Data Generation** We generate datasets $\mathcal{D} = \{(d_i, x^\star(d_i)\}_{i=1}^{1,000}$ for $k$-by-$k$ grids where $k \in \{5, 10, 20, 30, 50, 100\}$ and the $d_i$ are sampled uniformly at random from $[0, 1]^5$, the true edge weights are computed as $w(d) = Wd$ for fixed $W \in \mathbb{R}^{|\mathcal{E}| \times 5}$, and $x^\star(d)$ is computed given $w(d)$ using the aforementioned `pyTorch` implementation of Dijkstra's algorithm. Thus, all entries of $w(d)$ are non-negative, and are positive with overwhelming probability.

---

[4]adapted from `Tensorflow` code available at https://github.com/google-research/google-research/blob/master/perturbations/experiments/shortest_path.py

| grid size | number of variables | neural network size |
|:---:|:---:|:---:|
| 5-by-5 | 40 | 500 |
| 10-by-10 | 180 | 2040 |
| 20-by-20 | 760 | 8420 |
| 30-by-30 | 1740 | 19200 |
| 50-by-50 | 4900 | 53960 |
| 100-by-100 | 19800 | 217860 |

Table 1: Number of variables (*i.e.* number of edges) per grid size for the shortest path prediction problem of Section 5.2. Third column: number of parameters in $w_\Theta(d)$ for `DYS-Net`, `CVX-net` and `PertOpt-net`. For `BB-Net`, we found a latent dimension that is 20-times larger than the aforementioned three to be more effective.

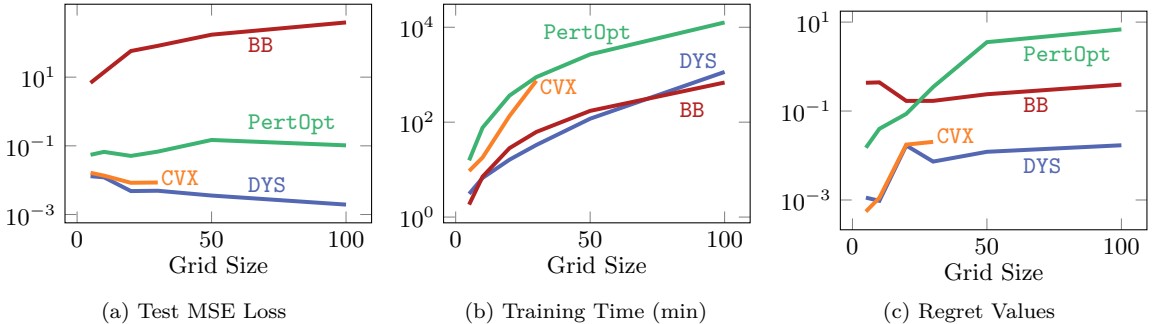

(a) Test MSE Loss      (b) Training Time (min)      (c) Regret Values

Figure 4: Results for the shortest path prediction problem. a) Test MSE loss (left), b) training time in minutes (middle), and c) regret values (right) vs. gridsize for DYS-Net (proposed) and approaches using `cvxpylayers` (Agrawal et al., 2019b) labeled CVX; Perturbed Optimization (Berthet et al., 2020) labeled PertOpt; and Blackbox Backpropagation (Vlastelica et al., 2019), labeled BB. Note CVX is unable to load or run problems with gridsize over 30. Dimensions of the variables can be found in Table 1.

**Models and Training** We test the same four approaches as in Sections 5.1.1 and 5.1.2, but unlike in these sections we do not use the `PyEPO` implementations of `PertOpt-net` or `BB-net`. We cannot, as the `PyEPO` implementations call `Gurobi` to solve equation 26 which, as mentioned above, cannot handle grids larger than $30 \times 30$. Instead, we use custom implementations based on the code accompanying the works introducing `PertOpt` (Berthet et al., 2020) and `BB` (Vlastelica et al., 2019). We use the same architecture for $w_\Theta(d)$ for `DYS-net`, `PertOpt-net`, and `Cvx-net`; a two layer fully connected neural network with leaky ReLU activation functions. For `DYS-net` and `Cvx-net` we *do not* modify the forward pass at test time. For `BB-net` we use a larger network by making the latent dimension 20-times larger than that of the first three as we found this more effective. Network sizes can be seen in Table 1.

We tuned the hyperparameters for each approach to the best of our ability on the smallest problem (5-by-5 grid) and then used these hyperparameter values for all other graph sizes. See Figure 3 for the results of this training run. We train all approaches for 100 epochs total on each problem using the $\ell_2$ loss equation 5.

**Results** The results are displayed in Figure 4. While `CVX-net` and `PertOpt-net` achieve low regret for small grids, `DYS-net` model achieves a low regret *for all* grids. In addition to training faster, `DYS-net` can also be trained for much larger problems, *e.g.*, 100-by-100 grids, as shown in Figure 4. We found that `CVX-net` could not handle grids larger than 30-by-30, *i.e.*, problems with more than 1740 variables[5] (see Table 1). Importantly, `PertOpt-net` takes close to a week to train for the 100-by-100 problem, whereas `DYS-net` takes about a day (see right Figure 4b). On the other hand, the training speed of `BB-net` is comparable to that of `DYS-net`, but does not lead to competitive accuracy as shown in Figure 4(a). Interpreting the outputs of `DYS-net` and `CVX-net` as (unnormalized) probabilities over the grid, one can use a greedy algorithm to determine the most probable path from top-left to bottom-right. For small grids, *e.g.* 5-by-5, this path coincides exactly with the true path for most $d$ (see Fig. 3).

### 5.3 Warcraft shortest path prediction

Finally, as an illustrative example, we consider the Warcraft terrains dataset first studied in Vlastelica et al. (2019). As shown in Figure 1, $d$ is a 96-by-96 RGB image, divided into a 12-by-12 grid of terrain tiles. Each tile has a different cost to traverse, and the goal is to find the quickest path from the top-left corner to the bottom-right corner.

**Models, Training, and Evaluation** We build upon the code provided as part of the `PyEPO` package (Tang & Khalil, 2022). We use the same architecture for $w_\Theta(d)$ for `BB-net`, `PertOpt-net`, and `DYS-net`—a truncated ResNet18 (He et al., 2016) as first proposed in Vlastelica et al. (2019). We train each network for 50 epochs, except for the baseline. The initial learning rate is $5 \times 10^{-4}$ and it is decreased by a factor of 10 after 30 and 40 epochs respectively. `PertOpt-net` and `DYS-net` are trained to minimize the $\ell_2$ loss equation 5, while `BB-net` is trained to minimize the Hamming loss as described in Vlastelica et al. (2019). We evaluate the quality of $w_\Theta(d)$ using normalized regret equation 25.

**Results** The results for this experiment are shown in Table 2. Interestingly, `BB-net` and `PertOpt-net` are much more competitive in this experiment than in the experiments of Sections 5.1.1, 5.1.2, and 5.2. We attribute this to the "discrete" nature of the true cost vector $w(d)$—for the Warcraft problem entries of $w(d)$ can only take on four, well-separated values—as opposed to the more "continuous" nature of $w(d)$ in the previous experiments. Thus, an algorithm that learns a rough approximation to the true cost vector will be more successful in the Warcraft problem than in the shortest path problems of Sections 5.1.1 and 5.2. This difference is illustrated in Figure 5. We also note that model selection using a validation set is beneficial for all approaches, but particularly so for `BB-net` and `DYS-net`, which achieve their best-performing models in the first several epochs of training.

---

[5]This is to be expected, as discussed in in Amos & Kolter (2017); Agrawal et al. (2019a)

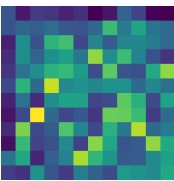 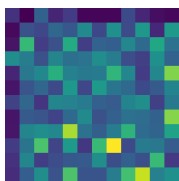 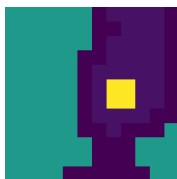 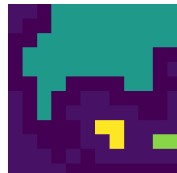

Figure 5: **Left two figures:** Sample cost matrices for shortest problem considered in Section 5.1.1. **Right two figures:** Sample cost matrices for the Warcraft shortest path prediction problem considered in Section 5.3. Note that in Section 5.3 the node weighted shortest path prediction problem is considered, while in Section 5.1.1 the edge weighted variant is solved. For ease of comparison, in the left two figures we have reshaped the edge cost vector into a node cost matrix.

| Algorithm | Test Normalized Regret | Time (in hours) |
|---|---|---|
| BB-net | 0.1204 | 0.11 |
| PertOpt-net | 0.1089 | 2.07 |
| DYS-net | 0.0889 | 0.09 |
| CVX-net | 0.0214 | 2.33 |

Table 2: Results for the 12-by-12 Warcraft shortest path prediction problem. We select the model achieving best normalized regret on the validation set. The time displayed is the time till this best normalized regret is achieved. All results are averaged over three runs.

## 6    Conclusions

This work presents a new method for learning to solve ILPs using Davis-Yin splitting which we call `DYS-net`. We prove that the gradient approximation computed in the backward pass of `DYS-net` is indeed a descent direction, thus advancing the current understanding of implicit networks. Our experiments show that `DYS-net` is capable of scaling to truly large ILPs, and outperforms existing state-of-the-art methods, at least when training data is of the form $(d, x^\star(d))$. We note that in principle `DYS-net` may be applied given training data of the form $(d, w(d))$; a common paradigm for many predict-then-optimize problems Mandi et al. (2023). However, preliminary experiments showed that `SPO+` Elmachtoub & Grigas (2022) achieved a substantially lower regret than `DYS-net` in this setting. We leave the extension of `DYS-net` to this data type to future work. Our experiments also reveal an interesting dichotomy between problems in which entries of $w(d)$ may take on only a handful of discrete values and problems in which $w(d)$ is more "continuous". Future work could explore this dichotomy further, as well as apply `DYS-net` to additional ILPs, for example the traveling salesman problem.

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

# A   Appendix A: Proofs

For the reader's convenience we restate each result given in the main text before proving it.

**Theorem 2.**   *Let $\mathcal{C}_1, \mathcal{C}_2$ be as in equation 14, and suppose $f_\Theta(x; \gamma, d) = w_\Theta(d)^\top x + \frac{\gamma}{2}\|x\|_2^2$ for any neural network $w_\Theta(d)$. For any $\alpha \in (0, 2/\gamma)$ define the sequence $\{z^k\}$ by:*

$$z^{k+1} = T_\Theta(z^k) \quad \text{for all } k \in \mathbb{N} \tag{28}$$

*where*

$$T_\Theta(z) \triangleq z - P_{\mathcal{C}_2}(z) + P_{\mathcal{C}_1}\left((2 - \alpha\gamma)\, P_{\mathcal{C}_2}(z) - z - \alpha w_\Theta(d)\right). \tag{29}$$

*If $x^k \triangleq P_{\mathcal{C}_2}(z^k)$ then the sequence $\{x^k\}$ converges to $x_\Theta$ in equation 8 and $\|x^{k+1} - x^k\|_2^2 = O(1/k)$.*

*Proof.* First note that $\nabla_x f_\Theta(x_\Theta; \gamma, d) = w_\Theta(d) + \gamma x$. Because $f_\Theta(x_\Theta; \gamma, d)$ is strongly convex, $x_\Theta$ is unique and is characterized by the first order optimality condition:

$$\nabla_x f_\Theta(x_\Theta; d)^\top (x - x_\Theta) \geq 0 \quad \text{for all } x \in \mathcal{C}. \tag{30}$$

Note that equation 30 can equally be viewed as a variational inequality with operator $F_\Theta(x; d) = \nabla_x f_\Theta(x_\Theta; d)$. We deduce that $z_\Theta$ with $x_\Theta = P_{\mathcal{C}_2}(z_\Theta)$ is a fixed point of

$$T_\Theta(z) \triangleq z - P_{\mathcal{C}_2}(z) + P_{\mathcal{C}_1}\left(2 \cdot P_{\mathcal{C}_2}(z) - z - \alpha\left[w_\Theta(d) + \gamma P_{\mathcal{C}_2}(z)\right]\right) \tag{31a}$$

$$= z - P_{\mathcal{C}_2}(z) + P_{\mathcal{C}_1}\left((2 - \alpha\gamma) \cdot P_{\mathcal{C}_2}(z) - z - \alpha w_\Theta(d)\right) \tag{31b}$$

from McKenzie et al. (2021, Theorem 4.2), which is itself a standard application of ideas from operator splitting (Davis & Yin, 2017; Ryu & Yin, 2022).

As $\nabla_x f_\Theta(x_\Theta; \gamma, d)$ is $\gamma$-Lipschitz continuous it is also $\nabla_x f_\Theta$ is $1/\gamma$-cocoercive by the Baillon-Haddad theorem (Baillon & Haddad, 1977; Bauschke & Combettes, 2009). McKenzie et al. (2021, Theorem 3.3) then implies that $z^k \to z_\Theta$ and $x^k \to x_\Theta$. However, this theorem does not give a convergence rate. In fact, the convergence rate can be deduced from known results; because $T_\Theta$ is averaged for $\alpha < 2/\gamma$ the rate $\|z^{k+1} - z^k\| = \mathcal{O}(1/k)$ follows from Ryu & Yin (2022, Theorem 1).   Thus,

$$\|x^{k+1} - x^k\| = \|P_{\mathcal{C}_2}(z^{k+1}) - P_{\mathbb{C}_1}(z^k)\| \leq \|z^{k+1} - z^k\| = \mathcal{O}(1/k), \tag{32}$$

as desired.

$\square$

Next, we state two auxiliary lemmas relating the Jacobian matrices to projections onto linear subspaces.

**Lemma 8.**   *If $\mathcal{C}_1 \triangleq \{x : Ax = b\}$, for full-rank $A \in \mathbb{R}^{m \times n}$, $b \in \mathbb{R}^n$ with $m < n$, and $\mathcal{H}_1 \triangleq \mathrm{Null}(A)$ then*

$$\frac{\partial P_{\mathcal{C}_1}}{\partial z} = P_{\mathcal{H}_1}, \quad \text{for all } z \in \mathbb{R}^n. \tag{33}$$

*Proof.* Let $A = U\Sigma V^\top$ denote the reduced SVD of $A$, and note that as $A \in \mathbb{R}^{m \times n}$ with $m < n$ we have $U \in \mathbb{R}^{m \times m}, \Sigma \in \mathbb{R}^{m \times m}$ and $V \in \mathbb{R}^{n \times m}$. Differentiating the formula for $P_{\mathcal{C}_1}$ given in Lemma 1 yields

$$\frac{\partial P_{\mathcal{C}_1}}{\partial z} = I - A^\dagger A, \tag{34}$$

where $A^\dagger \triangleq V\Sigma^{-1}U^\top$. Note

$$A^\dagger A = \left(V\Sigma^{-1}U^\top\right)\left(U\Sigma V^\top\right) = VV^\top, \tag{35}$$

which is the orthogonal projection onto $\mathrm{Range}(V) = \mathrm{Range}(A^\top)$. It follows that $I - A^\dagger A$ is the orthogonal projection on to $\mathrm{Range}(A^\top)^\perp = \mathrm{Null}(A)$.   $\square$

**Lemma 9.** *Define the multi-valued function*

$$c(\alpha) \triangleq \partial \max(0, \alpha) = \begin{cases} 1 & \text{if } \alpha > 0 \\ 0 & \text{if } \alpha < 0 \\ [0, 1] & \text{if } \alpha = 0 \end{cases} \tag{36}$$

*and, for $z \in \mathbb{R}^n$, define $\mathcal{H}_{2,z} \triangleq \operatorname{Span}(e_i : z_i > 0)$. Then*

$$\partial P_{\mathcal{C}_2}(\bar{z}) = \left[\frac{d}{dz} \operatorname{ReLU}(z)\right]_{z=\bar{z}} = \operatorname{diag}(c(\bar{z})), \tag{37}$$

*and adopting the convention for choosing an element of $\partial P_{\mathcal{C}_2}(\bar{z})$ stated in the main text:*

$$\left.\frac{dP_{\mathcal{C}_2}}{dz}\right|_{z=\bar{z}} = \operatorname{diag}(\tilde{c}(\bar{z})) = P_{\mathcal{H}_{2,\bar{z}}}. \tag{38}$$

*Proof.* First, suppose $\bar{z} \in \mathbb{R}^n$ satisfies $\bar{z}_i \neq 0$, for all $i \in [n]$, *i.e.* $\bar{z}$ is a smooth point of $P_{\mathcal{C}_2}$. Note

$$\frac{d[\operatorname{ReLU}(\bar{z}_i)]}{dz} = 1 \text{ if } i = j \text{ and } z_i > 0 \qquad \text{and} \qquad \frac{d[\operatorname{ReLU}(\bar{z}_i)]}{dz} = 0 \text{ if } i \neq j \text{ or } z_i < 0. \tag{39}$$

Thus, the Jacobian matrix is diagonal with a 1 in the $(i, i)$-th position whenever $\bar{z}_i > 0$ and 0 otherwise, *i.e.* $\left.\frac{dP_{\mathcal{C}_2}}{dz}\right|_{z=\bar{z}} = \operatorname{diag}(c(\bar{z}))$. Now suppose $z_i = 0$ for one $i$. For all $z \in \mathbb{R}^n$ with $z_i < 0$, the Jacobian $\left.\frac{dP_{\mathcal{C}_2}}{dz}\right|_z$ is well-defined and has a 0 in the $(i, i)$-th position, while for $z \in \mathbb{R}^n$ with $z_i > 0$, the Jacobian $\left.\frac{dP_{\mathcal{C}_2}}{dz}\right|_z$ is well-defined and has a 1 in the $(i, i)$-th position. Taking the convex hull yields the interval $[0, 1]$ in the $(i, i)$-th position, as claimed. The case where $\bar{z}_i = 0$ for multiple $i$ is similar.

Consequently, the product of $\left.\frac{dP_{\mathcal{C}_2}}{dz}\right|_{z=\bar{z}}$ and any vector $v \in \mathbb{R}^n$ equals $v$ if and only if $v \in \operatorname{Span}(e_i : \bar{z}_i > 0)$. This shows the linear operator is idempotent with fixed point set $\mathcal{H}_{2,\bar{z}}$, *i.e.* it is the projection operator $P_{\mathcal{H}_{2,\bar{z}}}$. $\qquad\square$

**Theorem 4.** *Suppose $A$ is full-rank and $\mathcal{H}_1 \triangleq \operatorname{Null}(A)$, $\mathcal{H}_{2,z} \triangleq \operatorname{Span}(e_i : z_i > 0)$. Then for all $\hat{z} \in \mathbb{R}^n$*

$$\left.\frac{\partial T_\Theta}{\partial z}\right|_{z=\bar{z}} = P_{\mathcal{H}_1^\perp} P_{\mathcal{H}_{2,\bar{z}}^\perp} + (1 - \alpha\gamma) \cdot P_{\mathcal{H}_1} P_{\mathcal{H}_{2,\bar{z}}}. \tag{40}$$

*Proof.* Differentiating the expression for $T_\Theta$ in equation 16 with respect to $z$ yields

$$\left.\frac{\partial T_\Theta}{\partial z}\right|_{z=\hat{z}} = I - \left.\frac{dP_{\mathcal{C}_2}}{dz}\right|_{z=\hat{z}} + \left.\frac{dP_{\mathcal{C}_1}}{dz}\right|_{z=y(\hat{z})} \left[(2 - \alpha\gamma) \cdot \left.\frac{dP_{\mathcal{C}_2}}{dz}\right|_{z=\hat{z}} - I\right] \tag{41a}$$

$$= I - P_{\mathcal{H}_{2,\hat{z}}} + P_{\mathcal{H}_1} \left((2 - \alpha\gamma)P_{\mathcal{H}_{2,\hat{z}}} - I\right), \quad \text{for all } \hat{z} \in \mathbb{R}^n, \tag{41b}$$

where, for notational brevity, we set $y(\hat{z}) \triangleq (2 - \alpha\gamma) \cdot P_{\mathcal{C}_2}(\hat{z}) - \hat{z} - \alpha w_\Theta(d)$ in the first line and the second line follows from Lemmas 8 and 9. Repeatedly using the fact, for any subspace $\mathcal{H} \subset \mathbb{R}^n$, $P_{\mathcal{H}^\perp} = I - P_{\mathcal{H}}$, the derivative $\partial T_\Theta / \partial z$ can be further rewritten:

$$\left.\frac{\partial T_\Theta}{\partial z}\right|_{z=\hat{z}} = I - P_{\mathcal{H}_{2,\hat{z}}} + (2 - \alpha\gamma) \cdot P_{\mathcal{H}_1} P_{\mathcal{H}_{2,\hat{z}}} - P_{\mathcal{H}_1} \tag{42a}$$

$$= P_{\mathcal{H}_{2,\hat{z}}^\perp} + (2 - \alpha\gamma) \cdot P_{\mathcal{H}_1} \left(I - P_{\mathcal{H}_{2,\hat{z}}^\perp}\right) - P_{\mathcal{H}_1} \tag{42b}$$

$$= P_{\mathcal{H}_{2,\hat{z}}^\perp} + P_{\mathcal{H}_1} + (1 - \alpha\gamma) \cdot P_{\mathcal{H}_1} - P_{\mathcal{H}_1} P_{\mathcal{H}_{2,\hat{z}}^\perp} - (1 - \alpha\gamma) \cdot P_{\mathcal{H}_1} P_{\mathcal{H}_{2,\hat{z}}^\perp} - P_{\mathcal{H}_1} \tag{42c}$$

$$= (I - P_{\mathcal{H}_1})P_{\mathcal{H}_{2,\hat{z}}^\top} + (1 - \alpha\gamma) \cdot P_{\mathcal{H}_1}(I - P_{\mathcal{H}_{2,\hat{z}}^\perp}) \tag{42d}$$

$$= P_{\mathcal{H}_1^\perp} P_{\mathcal{H}_{2,\hat{z}}^\perp} + (1 - \alpha\gamma) \cdot P_{\mathcal{H}_1} P_{\mathcal{H}_{2,\hat{z}}}, \quad \text{for all } \hat{z} \in \mathbb{R}^n, \tag{42e}$$

completing the proof. $\qquad\square$

We use the following lemma to prove Theorem 6.

**Lemma 10.** *If the LICQ condition holds at $x_\Theta$, then $\mathcal{H}_1^\perp \cap \mathcal{H}_{2,z_\Theta}^\perp = \{0\}$.*

*Proof.* We first rewrite $\mathcal{H}_1^\perp$ and $\mathcal{H}_{2,z_\Theta}^\perp$. The subspace $\mathcal{H}_{2,z_\Theta}^\perp$ is spanned by all non-positive coordinates of $z_\Theta$. By equation 17, $[x_\Theta]_i = \max\{0, [z_\Theta]_i\}$, and so $i \in \mathcal{A}(x_\Theta)$ if and only if $[z_\Theta]_i \leq 0$. It follows that

$$\mathcal{H}_{2,z_\Theta}^\perp \triangleq \mathrm{Span}\{e_i : [z_\Theta]_i \leq 0\} = \mathrm{Span}\{e_i : i \in \mathcal{A}(x_\Theta)\} = \mathrm{Span}\{e_{i_1}, \ldots, e_{i_\ell}\}, \tag{43}$$

where we enumerate $\mathcal{A}(x_\Theta)$ via $\mathcal{A}(x_\Theta) = \{i_1, \ldots, i_\ell\}$. On the other hand, $\mathcal{H}_1^\perp = \mathrm{Range}(A^\top) = \mathrm{Span}(a_1, \ldots, a_m)$ where $a_i^\top$ denotes the $i$-th row of $A$.

Let $v \in \mathcal{H}_1^\perp \cap \mathcal{H}_{2,z_\Theta}^\perp$ be given. Since $v \in \mathcal{H}_1^\perp$, there are scalars $\alpha_1, \ldots, \alpha_\ell$ such that $v = \alpha_1 e_{i_1} + \cdots + \alpha_\ell e_{i_\ell}$. Similarly, since $v \in \mathcal{H}_{2,z_\Theta}^\perp$, there are scalars $\beta_1, \ldots, \beta_m$ such that $v = \beta_1 a_1 + \cdots + \beta_m a_m$. Hence

$$0 = v - v = \left(\alpha_1 e_{i_1} + \ldots + \alpha_\ell e_{i_\ell}\right) - \left(\beta_1 a_1 + \ldots + \beta_m a_m\right). \tag{44}$$

By the LICQ condition, $\{e_{i_1}, \ldots, e_{i_\ell}\} \cup \{a_1, \ldots, a_m\}$ is a linearly independent set of vectors; hence $\alpha_1 = \ldots = \alpha_\ell = \beta_1 = \ldots = \beta_m = 0$ and, thus, $v = 0$. Since $v$ was arbitrarily chosen in $\mathcal{H}_1^\perp \cap \mathcal{H}_{2,z_\Theta}^\perp$, the result follows. $\square$

**Theorem 6.** *If the LICQ condition holds at $x_\Theta$, $A$ is full-rank and $\alpha \in (0, 2/\gamma)$, then $\|\partial T_\Theta / \partial z\|_{z=z_\Theta} < 1$.*

*Proof.* By Lemma 10, the LICQ condition implies $\mathcal{H}_1^\perp \cap \mathcal{H}_{2,z_\Theta}^\perp = \{0\}$. This implies that either (i) the first principal angle $\tau$ between these two subspaces is nonzero, and so the cosine of this angle is less than unity, *i.e.*

$$1 > \cos(\tau) \triangleq \max_{u \in \mathcal{H}_1^\perp : \|u\|=1} \max_{v \in \mathcal{H}_{2,z}^\perp : \|v\|=1} \langle u, v \rangle, \tag{45}$$

or (ii) (at least) one of $\mathcal{H}_1^\perp, \mathcal{H}_{2,z_\Theta}^\perp$ is the trivial vector space $\{0\}$. In either case, let $w \in \mathbb{R}^n$ be given. By Theorem 4, in case (ii)

$$\left[\frac{\partial T_\Theta}{\partial z} w\right]_{z=z_\Theta} = P_{\mathcal{H}_1^\perp} P_{\mathcal{H}_{2,z_\Theta}^\perp} w + (1 - \alpha\gamma) \cdot P_{\mathcal{H}_1} P_{\mathcal{H}_{2,z_\Theta}} w = (1 - \alpha\gamma) \cdot P_{\mathcal{H}_1} P_{\mathcal{H}_{2,z_\Theta}} w \tag{46}$$

implying that

$$\left\|\frac{\partial T_\Theta}{\partial z} w\right\|_{z=z_\Theta} = (1 - \alpha\gamma) \left\|P_{\mathcal{H}_1} P_{\mathcal{H}_{2,z_\Theta}} w\right\| \leq (1 - \alpha\gamma)\|w\|, \tag{47}$$

where the inequality follows as projection operators are firmly nonexpansive. In case (i), write $w = w_1 + w_2$, where $w_1 \in \mathcal{H}_{2,z_\Theta}$ and $w_2 \in \mathcal{H}_{2,z_\Theta}^\perp$. Appealing to Theorem 4 again

$$\left[\frac{\partial T_\Theta}{\partial z} w\right]_{z=z_\Theta} = P_{\mathcal{H}_1^\perp} P_{\mathcal{H}_{2,z_\Theta}^\perp} w + (1 - \alpha\gamma) \cdot P_{\mathcal{H}_1} P_{\mathcal{H}_{2,z_\Theta}} w = P_{\mathcal{H}_1^\perp} w_2 + (1 - \alpha\gamma) \cdot P_{\mathcal{H}_1} w_1. \tag{48}$$

Pythagoras' theorem may be applied to deduce, together with the fact $P_{\mathcal{H}_1^\perp} w_2$ and $P_{\mathcal{H}_1} w_1$ are orthogonal,

$$\left\|\frac{\partial T_\Theta}{\partial z} w\right\|_{z=z_\Theta}^2 = \left\|P_{\mathcal{H}_1^\perp} w_2\right\|^2 + (1 - \alpha\gamma)^2 \cdot \|P_{\mathcal{H}_1} w_1\|^2. \tag{49}$$

Since $w_2 \in \mathcal{H}_{2,z_\Theta}^\perp$, the angle condition (45) implies

$$\left\|P_{\mathcal{H}_1^\perp} w_2\right\|^2 = \langle P_{\mathcal{H}_1^\perp} w_2, P_{\mathcal{H}_1^\perp} w_2 \rangle = \langle w_2, P_{\mathcal{H}_1^\perp} P_{\mathcal{H}_1^\perp} w_2 \rangle = \langle w_2, P_{\mathcal{H}_1^\perp} w_2 \rangle \leq \cos(\tau) \cdot \|w_2\|^2, \tag{50}$$

where the third equality holds since orthogonal linear projections are symmetric and idempotent. Because projections are non-expansive and $P_{\mathcal{H}_{2,z_\Theta}}$ is linear,

$$\|P_{\mathcal{H}_1} w_1\|^2 = \|P_{\mathcal{H}_1} w_1 - P_{\mathcal{H}_1} 0\|^2 \leq \|w_1 - 0\|^2 = \|w_1\|^2. \tag{51}$$

Combining (49), (50) and (51) reveals

$$\left\| \frac{\partial T_\Theta}{\partial z} w \right\|_{z=z_\Theta}^2 \leq \cos(\tau) \cdot \|w_2\|^2 + (1 - \alpha\gamma)^2 \|w_1\|^2 \tag{52a}$$

$$\leq \max\{\cos(\tau), (1 - \alpha\gamma)^2\} \cdot \left( \|w_1\|^2 + \|w_2\|^2 \right) \tag{52b}$$

$$= \max\{\cos(\tau), (1 - \alpha\gamma)^2\} \cdot \|w\|^2, \tag{52c}$$

noting the final equality holds since $w_1$ and $w_2$ are orthogonal. Because (52) holds for arbitrarily chosen $w \in \mathbb{R}^n$,

$$\left\| \frac{\partial T_\Theta}{\partial z} \right\|_{z=z_\Theta} \triangleq \sup \left\{ \left\| \frac{\partial T_\Theta}{\partial z} w \right\|_{z=z_\Theta} : \|w\| = 1 \right\} \leq \sqrt{\max\{\cos(\tau), (1 - \alpha\gamma)^2\}} < 1, \tag{53}$$

where the final inequality holds by (45) and the fact $\alpha \in (0, 2/\gamma)$ implies $1 - \alpha\gamma \in (-1, 1)$, as desired. $\qquad\square$

**Corollary 7.** *If $T_\Theta$ is continuously differentiable with respect to $\Theta$ at $z_\Theta$, the assumptions in Theorem 6 hold and $(\partial T_\Theta/\partial \Theta)^\top (\partial T_\Theta/\partial \Theta)$ has condition number sufficiently small, then $p_\Theta(x_\Theta)$ is a descent direction for $\ell$ with respect to $\Theta$.*

*Proof.* From the proof of Theorem 6 we see that $T_\Theta$ is contractive with constant $\Gamma = \sqrt{\max\{\cos(\tau), (1 - \alpha\gamma)^2\}}$ and so the main theorem of (Fung et al., 2022), guaranteeing $p_\Theta$ is a descent direction, as long as the condition number of $(\partial T_\Theta/\partial \Theta)^\top (\partial T_\Theta/\partial \Theta)$ is less than $1/\Gamma$. $\qquad\square$

**Remark 11.** *Similar guarantees, albeit with less restrictive assumptions on $\partial T_\Theta/\partial \Theta$, can be deduced from the results of the recent work (Bolte et al., 2024).*

## B  Derivation for Canonical Form of Knapsack Prediction Problem

For completeness, we explain how to transform the $k$-knapsack prediction problem into the canonical form equation 8, and how to derive the standardized representation of the constraint polytope $\mathcal{C}$. Recall that the $k$-knapsack prediction problem, as originally stated, is

$$x^\star = \arg\max_{x \in \mathcal{X}} w^\top x \text{ where } \mathcal{X} = \{x \in \{0,1\}^\ell : \ Sx \leq c\} \text{ and } S = \begin{bmatrix} s_1 & \cdots & s_\ell \end{bmatrix} \in \mathbb{R}^{k \times \ell} \tag{54}$$

We introduce slack variables $y_1, \ldots, y_k$ so that the inequality constraint $Sx \leq c$ becomes

$$-Sx + c \geq 0 \implies -Sx + c = y \text{ and } y \geq 0$$

$$\implies \begin{bmatrix} S & I_k \end{bmatrix} \begin{bmatrix} x \\ y \end{bmatrix} = c$$

We relax the binary constraint $x_i \in \{0,1\}$ to $0 \leq x_i \leq 1$. We add additional slack variables $z_1, \ldots, z_\ell$ to account for the upper bound:

$$1 - x_i \geq 0 \implies 1 - x_i = z_i \text{ and } z_i \geq 0 \implies \begin{bmatrix} I_{\ell \times \ell} & 0_{\ell \times k} & I_{\ell \times \ell} \end{bmatrix} \begin{bmatrix} x \\ y \\ z \end{bmatrix} = \mathbf{1} \tag{55}$$

Putting this together, define

$$A = \begin{bmatrix} -S & -I_{k \times k} & 0_{k \times \ell} \\ I_{\ell \times \ell} & 0_{\ell \times k} & I_{\ell \times \ell} \end{bmatrix} \in \mathbb{R}^{(k+\ell) \times (2\ell+k)} \text{ and } b = \begin{bmatrix} -c \\ \mathbf{1}_\ell \end{bmatrix} \in \mathbb{R}^{k+\ell} \tag{56}$$

Finally, redefine $x = \begin{bmatrix} x & y & z \end{bmatrix}^\top$ and $w = \begin{bmatrix} -w & \mathbf{0}_k & \mathbf{0}_\ell \end{bmatrix}$ (where we're using $-w$ to switch the argmax to an argmin) and obtain:

$$x^\star = \arg\min_{x \in \text{Conv}(\mathcal{X})} w^\top(d)x + \gamma\|x\|_2^2 \text{ where } \text{Conv}(\mathcal{X}) = \{x : Ax = b \text{ and } x \geq 0\} \tag{57}$$

## C   Experimental Details

### C.1   Additional Data Details for `PyEPO` problems

We use PyEPO (Tang & Khalil, 2022), specifically the functions `data.shortestpath.genData` and `data.knapsack.genData` to generate the training data. Both functions sample $B \in \mathbb{R}^{n \times 5}$ with $B_{ij} = +1$ with probability 0.5 and $-1$ with probability 0.5 and then construct the cost vector $w(d)$ as

$$[w(d)]_i = \left[ \frac{1}{3.5^{\text{deg}}} \left( \frac{1}{\sqrt{5}} (Bd)_i + 3 \right)^{\text{deg}} + 1 \right] \cdot \epsilon_{ij}$$

where $\text{deg} = 4$ and $\epsilon_{ij}$ is sampled uniformly from the interval $[0.5, 1.5]$. Note that $w(d)$ is, by construction, non-negative.

### C.2   Additional Model Details for `PyEPO` problems

For `PertOpt-net` and `BBOpt-net` we use the default hyperparameter values provided by `PyEPO`, namely $\lambda = 5$ for `BBOpt-net`, number of samples equal 3, $\epsilon = 1$ and Gumbel noise for `PertOpt-net`. See Pogančić et al. (2019) and Berthet et al. (2020) respectively for descriptions of these hyperparameters. We do so as these are selected via hyperparameter tuning Tang & Khalil (2022). As `DYS-net` and `CVX-net` solve a regularized form of the underlying LP equation 7, the regularization parameter $\gamma$ needs to be set. After experimenting with different values, we select $\gamma = 5 \times 10^{-4}$ for `DYS-net` and $\gamma = 5 \times 10^{-1}$ for `CVX-net`.

### C.3   Additional Training Details for Knapsack Problem

For all models we use an initial learning rate of $10^{-3}$ and a scheduler that reduces the learning rate whenever the validation loss plateaus. We also used weight decay with a parameter of $5 \times 10^{-4}$.

### C.4   Additional Model Details for Large-Scale Shortest Path Problem

Our implementation of `PertOpt-net` used a PyTorch implementation[6] of the original TensorFlow code[7] associated to the paper Berthet et al. (2020). We experimented with various hyperparameter settings for 5-by-5 grids and found setting the number of samples equal to 3, the temperature (*i.e.* $\varepsilon$) to 1 and using Gumbel noise to work best, so we used these values for all other shortest path experiments. Our implementation of `BB-net` uses the `blackbox-backprop` package[8] associated to the paper Pogančić et al. (2019). We found setting $\lambda = 100$ to work best for 5-by-5 grids, so we use this value for all other shortest path experiments. Again, we select $\gamma = 5 \times 10^{-4}$ for `DYS-net` and $\gamma = 5 \times 10^{-1}$ for `CVX-net`

### C.5   Additional Training Details for Large-Scale Shortest Path Problem

We use the MSE loss to train `DYS-net`, `PertOpt-net`, and `CVX-net`. We tried using the MSE loss with `BB-net` but this did not work well, so we used the Hamming (also known as 0–1) loss, as done in Pogančić et al. (2019).

To train `DYS-net` and `CVX-net`, we use an initial learning rate of $10^{-2}$ and use a scheduler that reduces the learning rate whenever the test loss plateaus—we found this to perform the best for these two models. For `PertOpt-net` we found that using a fixed learning rate of $10^{-2}$ performed the best. For `BB-net`, we performed a logarithmic grid-search on the learning rate between $10{-1}$ to $10^{-4}$ and found that $10^{-3}$ performed best; we also attempted adaptive learning rate schemes such as reducing learning rates on plateau but did not obtain improved performance.

---

[6]See code at `github.com/tuero/perturbations-differential-pytorch`
[7]See code at `github.com/google-research/google-research/tree/master/perturbations`
[8]See code at `https://github.com/martius-lab/blackbox-backprop`

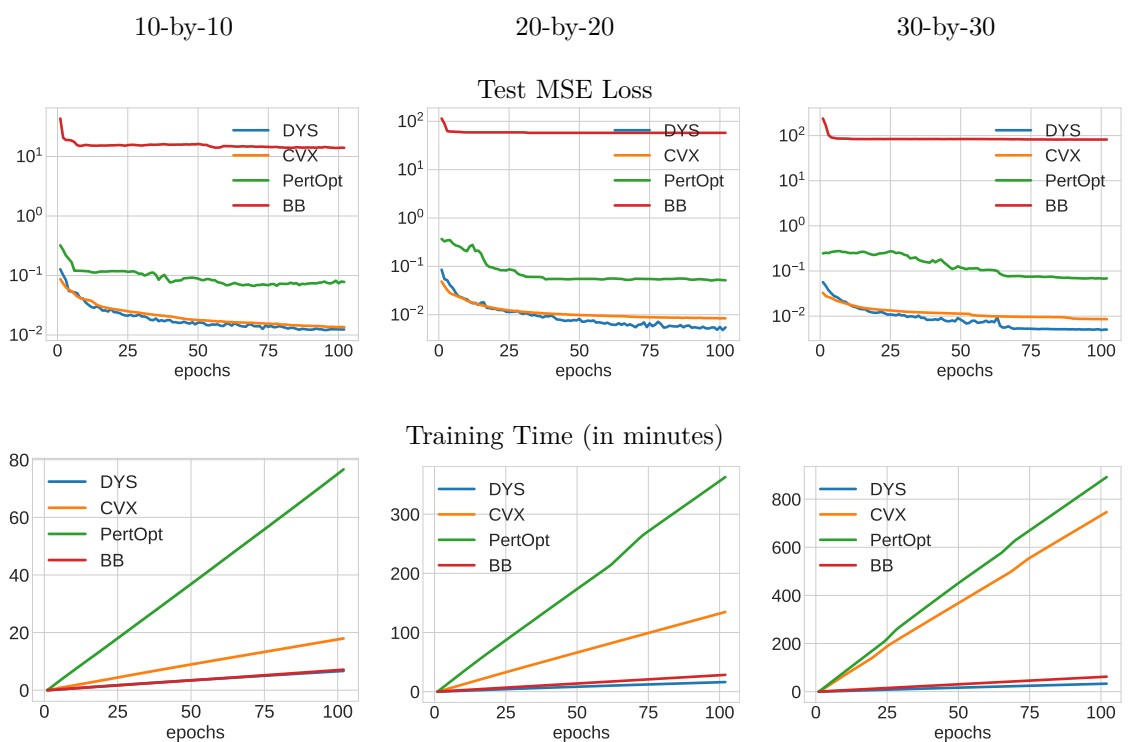

Figure 6: Comparison of of DYS-Net, `cvxpylayers` (Agrawal et al., 2019a), PertOptNet (Berthet et al., 2020), and Blackbox Backpropagation-net (BB-Net) (Pogančić et al., 2019) for three different grid sizes: $10 \times 10$ (first column), $20 \times 20$ (second column), and $30 \times 30$ (third column). The first row shows the MSE loss vs. epochs of the testing dataset. The second row shows the training time vs. epochs.

### C.6 Additional Model Details for Warcraft Experiment

For `PertOpt-net` and `BBOpt-net` we again use the default hyperparameter values provided by `PyEPO`, namely $\lambda = 10$ for `BBOpt-net`, number of samples equal 3, $\epsilon = 1$ and Gumbel noise for `PertOpt-net`. We set $\gamma = 0.5$ for both `CVX-net` and `DYS-net`.

### C.7 Hardware

All networks were trained using a AMD Threadripper Pro 3955WX: 16 cores, 3.90 GHz, 64 MB cache, PCIe 4.0 CPU and an NVIDIA RTX A6000 GPU.

## D Additional Experimental Results

In Figure 6, we show the test loss and training time per epoch for all three architectures: `DYS-net`, `CVX-net`, and `PertOpt-net` for 10-by-10, 20-by-20, and 30-by-30 grids. In terms of MSE loss, `CVX-net` and `DYS-net` lead to comparable performance. In the second row of Figure 6, we observe the benefits of combining the three-operator splitting with JFB (Fung et al., 2022); in particular, `DYS-net` trains much faster. Figure 7 shows some randomly selected outputs for the three architectures once fully trained.

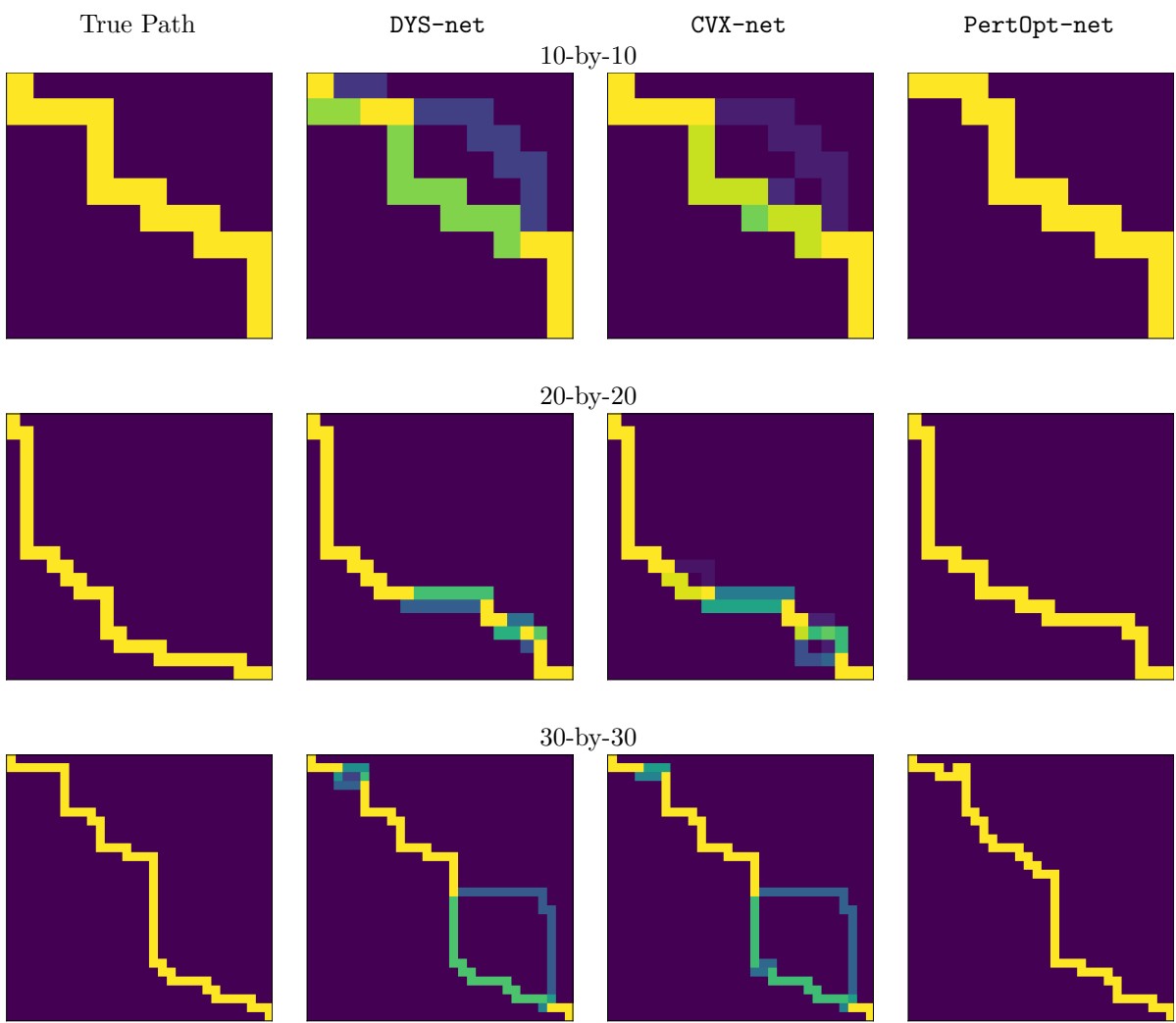

Figure 7: True paths (column 1), paths predicted by DYS-net (column 2), CVX-net (column 3), and PertOpt-net (column 4). Samples are taken from different grid sizes: 10-by-10 (row 1), 20-by-20 (row 2), and 30-by-30 (row 3).

