# OpenReview forum: "Differentiating Through Integer Linear Programs with Quadratic Regularization and Davis-Yin Splitting"
_TMLR — Accepted by TMLR_

### Review · Reviewer_DjLc · 2024-05-05

**Summary Of Contributions:**

The authors propose a method called `DYS-Net` for automatic differentiation of Integer Linear Programs (ILPs) with respect to a cost vector. The goal is to facilitate the use of ILPs in deep learning pipelines, especially when the dimension of the problem gets large.

To approximate the Jacobian of the ILP
$$
x(w) = \mathrm{argmin} \\{w^\top x : x \in \mathcal{C} \cap \mathbb{Z}^n \\}
\quad \text{where} \quad
\mathcal{C} = \\{x \in \mathbb{R}^n : Ax = b, x \geq 0 \\},
$$

`DYS-Net` combines known ideas from convex optimization and decision-focused learning in a novel way:

1. Relax the ILP and add a quadratic regularization to the objective, yielding a Linearly Constrained Quadratic Program (LCQP) [Wilder et al., 2019]
$$
x(w) = \mathrm{argmin} \\{w^\top x + \frac{\gamma}{2} \lVert x \rVert^2 : x \in \mathcal{C} \\}
$$
2. Solve the LCQP with a projected gradient algorithm, using Davis-Yin splitting to efficiently compute the projection onto the polynomial $\mathcal{C}$ [McKenzie et al., 2023]. The key ingredients are the decomposition
$$
\mathcal{C} = \mathcal{C}_1 \cup \mathcal{C}_2
\quad \text{where} \quad
\mathcal{C}_1 = \\{ x \in \mathbb{R}^n : Ax = b \\}
\quad \text{and} \quad
\mathcal{C}_2 = \\{ x \in \mathbb{R}^n : x \geq 0 \\}
$$
and the use of a precomputed SVD for the fixed constraint matrix $A$.
3. Avoid backpropagating through the iterative procedure, and instead leverage implicit differentiation on the fixed point condition of the projected gradient [Blondel et al., 2022]
4. Approximate the implicit function theorem by replacing the Jacobian of the fixed point condition with the identity [Fung et al, 2022]

A theorem justifies that the Jacobian-free heuristic finds a descent direction, which is non-trivial to obtain from previous results in the literature.
Numerical experiments showcase the efficiency and precision of `DYS-Net` compared with competing approaches on two standard benchmarks: the shortest path problem and the knapsack problem.

**Audience:**

Yes

**Broader Impact Concerns:**

There are no ethical implications that would require a broader impact statement.

**Claims And Evidence:**

No

**Requested Changes:**

## Important requests (necessary for acceptance)

- Clarify the performance impact of the SVD step, and whether the SVD is used in practice or only for proof purposes
- Discuss the possible use of iterative linear solvers in lieu of (pseudo)inversion
- Comment on what happens in realistic scenarios where the constraint matrix $A$ is hard to write down explicitly, but instead given implicitly through a domain-specific language (like [cvxpy](https://www.cvxpy.org/) or [JuMP](https://jump.dev/))
- Fix the crucial notation switches between $\mathcal{C}_1$ and $\mathcal{C}_2$ that occurred e.g. in Equation (19) and the one before.
- The paper mentions a "provided code" but I couldn't find it, did I just miss the line? Or was it removed due to anonymity concerns? If it is the latter, tools such as https://anonymous.4open.science/ allow for anonymous copies of existing GitHub repos
- Make sure that the cost vector $w(d)$ for both categories of experiments remains nonnegative. For instance in the shortest paths, $w(d) = Wd$ with $d_k \sim \mathcal{U}[0, 1]$ but the sign of $W$ is not specified, and the issue is similar for the knapsack (appendix C.1)

## Additional suggestions and questions

- Shorten the introductive part of the abstract, and put more focus on the actual contributions of the paper
- Could your framework be used when there are no "ground truth" optimal solutions $x^\star$, that is, when learning by experience instead of imitation?
- In your Theorem 2, the result $\lVert x^{k+1} - x^k \rVert_2^2 = O(1/k)$ does not seem to imply that the sequence converges. Is that truly what you wanted to write?
- In Corollary 7, how do you ensure that the condition number of $(\partial_{\Theta} T)^\top \partial_{\Theta} T$ is small enough? Is this something the user can influence? How do we check it?
- Ideally, keep the notations for $\mathcal{C}_1$ and $\mathcal{C}_2$ coherent with [McKenzie et al., 2023]: when I compare your equation (16) with their equation (4.5), they are reversed. This might be the cause of the switches mentioned above
- Fix the references to the theorems in [McKenzie et al., 2023]: I think what you call 3.2 and 3.3 are actually 4.2 and 4.3, and one of them is a Lemma in the published version
- In the proof of Theorem 6, can you give a reference for the result on the first principal angle between subspaces with zero intersection? It was not obvious to me intuitively, although I don't doubt it
- You claim that you cannot use PyEPO for grid sizes larger than 30 because it automatically calls Gurobi. However, you can actually plug any algorithm into PyEPO with a [user-defined model](https://khalil-research.github.io/PyEPO/build/html/content/examples/model.html#user-defined-models-from-scratch), so a Dijkstra implementation is also possible instead of Gurobi.
- I'm very surprised that CVX is unable to load when the grid size exceeds 30. Is the constraint matrix $A$ encoded in a sparse manner?
- Even if the cost vector given by $w(d)$ is nonnegative to begin with, the added random vector of `PertOpt` or the black box interpolation of `BB` might change $w$ so much that one of its components becomes negative. This is the reason behind the multiplicative perturbation introduced in [[Dalle et al., 2022](https://arxiv.org/abs/2207.13513)], which preserves the sign. Of course such a behavior is rarely observed if the perturbation is small enough, and the original code from Berthet et al implicitly handles it with (what I think is) an incorrect Dijkstra implementation. I'm not asking you to change your experiments because of that, but I would be curious to see what you think, and if it is worthy of discussion.

## Typos and style

- Page 2: "L20" (L twenty) $\to$ "L2O"
- Page 6: "Lipshitz" $\to$ "Lipschitz"
- Page 10, Table 1: "network size" $\to$ "neural network size" (not to be confused with the graph)
- Page 16, proof of Theorem 2: $\nabla_x f(x_{\Theta}; \gamma, d)$ $\to$ $\nabla_x f(x; \gamma, d)$

**Strengths And Weaknesses:**

## Strengths

**Would some individuals in TMLR's audience be interested in the findings of this paper? YES**

The paper is really well-written. It is easy to follow despite the intrication of several different ideas, and explains in detail the motivation for `DYS-Net`. I have enjoyed it very much, and will use it in my research once it is published.

The proposed method `DYS-Net` is straightforward to implement and fully GPU-compatible. It is designed to scale to large instances thanks to its use of projected gradient, instead of costly primal-dual methods that differentiate through KKT conditions.

The theoretical analysis is sound, and shows that the counter-intuitive Jacobian-free approach still works in this non-smooth, non-contractive case.

## Weaknesses

**Are the claims made in the submission supported by accurate, convincing and clear evidence? NOT YET**

My main criticism is that `DYS-Net` claims to allow high-dimensional settings... but still requires computing a full SVD for the constraint matrix $A$. The SVD is a very expensive (cubic) operation, so even if you just perform it once at the beginning, I'm unsure how much you gain on truly large instances.
Furthermore, the constraint matrix of real-life problems has a lot of structure (sparsity, blocks, bands) which the SVD does not exploit. Typically, in graph optimization, the constraint matrix is some variant of the adjacency matrix. And unfortunately, I think (but I'm not 100% sure) that the SVD of a sparse matrix is not sparse in general.

Luckily, I think there is a solution. Indeed, the SVD $A = U \Sigma V^\top$ is only used to derive the pseudoinverse $A^\dagger = V \Sigma^{-1} U^\top$, and the pseudoinverse is only used for solving linear systems in $P_{\mathcal{C}_1}(z) = z - A^\dagger (Az - b)$.
Given this observation, why not solve the linear system directly with the matrix $A$, using an iterative Krylov solver like GMRES which only requires matrix-vector products?
The same remark applies to the solution of a linear system involving the Jacobian $\mathcal{J}$: no need to compute a (pseudo)inverse if a Krylov solver is available.

Of course, the previous criticism is only valid if the SVD is actually used in the `DYS-Net` algorithm. If I understand correctly, the Jacobian-free approach dispenses us from ever computing $P_{\mathcal{C}_1}$, which means we never need the SVD in the code? I am not completely certain because the notations $\mathcal{C}_1$ and $\mathcal{C}_2$ are sometimes switched (probably typos).
In any case, if the SVD is only an ingredient of the theoretical proofs, then it must be clarified, and there is no need to insist so much on the performance of the split projection operator.

My other criticism concerns the experimental part. First, I am not sure that the encoder $w(d)$ is guaranteed to yield positive weights, which would be a problem both for shortest paths (with Dijkstra's algorithm in particular) and for knapsack.
Second, I do not understand how `DYS-Net` can be more precise than `CVX`. Faster, for sure, but as far as precision is concerned, `DYS-Net` can only be worse than the true convex program, right?

---

> ### Author Response · Authors · 2024-06-21
> **Response to Reviewer DjLc (part 1)**
>
> 1 _My main criticism is that DYS-Net claims to allow high-dimensional settings... but still requires computing a full SVD for the constraint matrix . The SVD is a very expensive (cubic) operation, so even if you just perform it once at the beginning, I'm unsure how much you gain on truly large instances. Luckily, I think there is a solution. Indeed, the SVD $A = U \Sigma V^\top$ is only used to derive the pseudoinverse $A^\dagger = V \Sigma^{-1} U^\top$, and the pseudoinverse is only used for solving linear systems in $P_{\mathcal{C}_1}(z) = z - A^\dagger (Az - b)$. Given this observation, why not solve the linear system directly with the matrix $A$, using an iterative Krylov solver like GMRES which only requires matrix-vector products?_
>
> **Response:** Thank you for this interesting observation! The computation of the SVD is a _once-off, offline, cost_. Using an iterative solver would be an _online cost_ incurred for each iteration of every forward pass for every sample of every epoch. So, it is not clear to us that this would indeed be cheaper. Moreover, this cost would be incurred once the trained model is deployed, whereas this is not the case with the SVD approach. We ahve added a remark below Lemma 1 discussing this.
>
> That said, you are correct that computing the SVD may be a computational bottleneck of `DYS-Net` for truly large problems, and future work could investigate ameliorating this. However, we maintain that `DYS-Net` scales to large problems better than existing baseline methods, as shown by our experimental results.
>
> 2. _Of course, the previous criticism is only valid if the SVD is actually used in the DYS-Net algorithm. If I understand correctly, the Jacobian-free approach dispenses us from ever computing $P_{\mathcal{C}_1}$, which means we never need the SVD in the code._
>
> **Response:** JFB does not circumvent having to compute $P_{\mathcal{C}_1}$. Instead, we avoid computation of the projection onto the intersection $\mathcal{C}_1 \cap \mathcal{C}_2$ by _combining JFB and DYS_. The use of Jacobian-Free Backpropagation also allows us to avoid computing the Jacobian $dPC_1 /dz$ in the backward pass.
>
> 4. _My other criticism concerns the experimental part. First, I am not sure that the encoder $w(d)$ is guaranteed to yield positive weights, which would be a problem both for shortest paths (with Dijkstra's algorithm in particular) and for knapsack_
>
> **Response:**
> For the experiments in Section 5.1 we use the Pyepo functions data.knapsack.genData and data.shortestpath.genData to generate $(d, w(d))$,
> and the associated solutions $x^\star_d$. We set the degree parameter to 4 (i.e. an even number) which guarantees that all entries of $w(d)$ are positive (see equation 8 in the arxiv version of Tang and Khalil (2022). For our “large scale” shortest path experiments of Section 5.2 we use $w(d) = W d$ where $W \in \mathbb{R}^{|E|×5}$ and $d \in \mathbb{R}^5$ have entries drawn uniformly from $[0, 1]$. Thus, all entries of $w(d)$ are non-negative, and are positive with overwhelming probability. Thanks for the comment, we have included these details in the manuscript.
>
> 5. _Second, I do not understand how DYS-Net can be more precise than CVX. Faster, for sure, but as far as precision is concerned, DYS-Net can only be worse than the true convex program, right?_
>
> **Response:** We believe your intuition here is mostly correct, but there are a few points to clarify:
>
>  - CVX doesn’t solve the “true” convex program, as in all cases in the paper this is a linear program (LP). In order to compute an informative gradient, using `CVX-net` and `DYS-net` we need to add a small quadratic regularizer so that the objective function is strongly convex. So, `CVX-net` and `DYS-net` are solving the same problem, see equations 7 and 8 in our submission.
>
>  - When training, we solve the regularized problem as described above, but at test time we solve the true LP using Gurobi. We discuss this in the “implementation” paragraph of Section 4 and in the “Models” paragraph of section 5.1, but will make this more clear.
>
>  That said, we find that CVX (or more precisely, the cvxpylayers package built upon CVX) works very well for small problems. When the performance of DYS-net begins to exceed CVX (see, e.g., Figure 2 (a)), it is likely because CVX is only able to train for a handful of epochs before reaching the 30 minute train time limit (see Figure 2 (b)), whereas DYS-net completes hundreds of epochs in the same amount of time.

---

> > ### Comment · Reviewer_DjLc · 2024-07-01
> >
> > > Response: Thank you for this interesting observation! The computation of the SVD is a once-off, offline, cost. Using an iterative solver would be an online cost incurred for each iteration of every forward pass for every sample of every epoch. So, it is not clear to us that this would indeed be cheaper. Moreover, this cost would be incurred once the trained model is deployed, whereas this is not the case with the SVD approach. We ahve added a remark below Lemma 1 discussing this.
> >
> > Thank you for these clarifications, in such cases I can see how the SVD makes sense.
> > However, it is important to keep in mind that for truly large instances, $A$ is typically sparse but its SVD will be dense, and likely won't fit in memory. This pitfall is avoided in your evaluations because the instances still have moderate size.
> >
> > > Response: JFB does not circumvent having to compute $P_{C_1}$. Instead, we avoid computation of the projection onto the intersection $C_1 \cap C_2$ by combining JFB and DYS. The use of Jacobian-Free Backpropagation also allows us to avoid computing the Jacobian $dP_{C_1} / dz$ in the backward pass.
> >
> > Let me rephrase my question: assume we replace the forward pass with any other GPU-compatible QP solver, as discussed with reviewer UhYG. Then, is $P_{C_1}$ ever used in the backward pass? And if not, does that mean we don't need any SVD?

---

> > > ### Author Response · Authors · 2024-07-02
> > > **Regarding computing $P_{C_1}$**
> > >
> > > **Response:** A minor clarification: $P_{C_1}$ is used in the forward pass, not the backward pass. We need to evaluate $P_{C_1}$ for each evaluation of $T_{\Theta}$, but $P_{C_1}$ is not used in computing the pseudogradient $p_{\Theta}$ (see eq. 19)
> > >
> > > By changing the solver on the forward pass, it may be possible to avoid calling $P_{C_1}$. For example, using OSQP [1] will avoid this. But, it will not avoid factoring large matrices, as an LDL decomposition of an $mn \times mn$ matrix needs to be computed, where $m$ is the number of constraints and $n$ is the number of variables.

---

> ### Author Response · Authors · 2024-06-21
> **Response to Reviewer DjLc (part 2)**
>
> -------
>
> #### Requested Changes
>
> 1. _Clarify the performance impact of the SVD step, and whether the SVD is used in practice or only for proof purposes_
>
> **Response:** As discussed above, the SVD is used in practice. Based on our numerical results, we do not think that the computation of the SVD results in an undue computational burden for problems with as many as 20, 000 variables. We have added some discussion on this point to the paper.
>
> 2. _Discuss the possible use of iterative linear solvers in lieu of (pseudo)inversion_
>
> **Response:** We refer to our response 1 in the point-by-point responses. We have also added a brief discussion of this to the paper.
>
> 3. _Comment on what happens in realistic scenarios where the constraint matrix $A$ is hard to write down explicitly, but instead given implicitly through a domain-specific language (like cvxpy or JuMP)
>
> **Response:** This is a great comment. If $A$ is given implicitly, then it would be most practical to build a pseudoinverse offline using an iterative scheme such as GMRES or Lanczos (see response 1 in point-by-point). We have added a brief discussion of this to the paper.
>
> 4. _Fix the crucial notation switches between  $C_1$ and $C_2$ that occurred e.g. in Equation (19) and the one before._
>
> **Response:** Thank you for catching this. We have fixed the notation switch.
>
> 5. _The paper mentions a "provided code" but I couldn't find it, did I just miss the line? Or was it removed due to anonymity concerns? If it is the latter, tools such as https://anonymous.4open.science/ allow for anonymous copies of existing GitHub repos_
>
> **Response:** The code is in a GitHub repo, the link was removed to preserve
> anonymity. The code is also provided as a zip file in the “supplementary material”
>
> 5. _Make sure that the cost vector $w(d)$ for both categories of experiments remains nonnegative. For instance in the shortest paths, $w(d) = Wd$ with $d_k \sim \mathcal{U}[0,1]$ but the sign of $W$ is not specified, and the issue is similar for the knapsack (appendix C.1)_
>
> **Response:** This is an important point; thanks for pointing it out! In all cases, $w(d)$ is indeed positive by construction, and we have made this clearer in the paper. See response 4 in the point-by-point.
>
> ------
>
> #### Additional suggestions and questions
>
> 1. _Shorten the introductive part of the abstract, and put more focus on the actual contributions of the paper_
>
> **Response:** Thank you, we have added more details on the actual contributions. In particular, we describe the use of Davis Yin splitting and Jacobian-free backpropagation.
>
> 2. _Could your framework be used when there are no "ground truth" optimal solutions $x^\star$, that is, when learning by experience instead of imitation?_
>
> **Response:** Yes, this can be done. For example one could use {\tt DYS-net} with regret as a loss function:
> \begin{equation}
>     \ell(\Theta;d_i) = w_i^{\top}x_{\Theta}(d_i)
> \end{equation}
> (assuming some access to the true cost vectors $w_i \approx w(d_i)$ at train time).
>
> 3. _In your Theorem 2, the result $\| x^{k+1} - x^k \| = \mathcal{O}(1/k)$ does not seem to imply that the sequence converges. Is that truly what you wanted to write?_
>
> **Response:** Thank you for pointing this out; this is an important point to clarify. Indeed, the sequence $\{x^k\}$ does  converge. We updated the wording of this theorem to note that $\{x^k\}$ converges and the residual converges at $\mathcal{O}(1/k)$ and cite the relevant work showing this in the proof.
>
> 4. _In Corollary 7, how do you ensure that the condition number of $(\partial_\Theta T)^\top (\partial_\Theta T)$ is small enough? Is this something the user can influence? How do we check it?_
>
> **Response:** This is an excellent point. Training has been observed to work well empirically in this work and other papers, but this is the primary gap between theory and practice in our work.
>
> 5. _Ideally, keep the notations for
>  and $\mathcal{C}_1$ and $\mathcal{C}_2$
>  coherent with [McKenzie et al., 2023]: when I compare your equation (16) with their equation (4.5), they are reversed. This might be the cause of the switches mentioned above_
>
> **Response:** This has been addressed above.
>
> 6. _Fix the references to the theorems in [McKenzie et al., 2023]: I think what you call 3.2 and 3.3 are actually 4.2 and 4.3, and one of them is a Lemma in the published version_
>
> **Response:** Thank you, we have fixed these references in the manuscript.

---

> > ### Comment · Reviewer_DjLc · 2024-07-01
> >
> > > Response: As discussed above, the SVD is used in practice. Based on our numerical results, we do not think that the computation of the SVD results in an undue computational burden for problems with as many as 20, 000 variables. We have added some discussion on this point to the paper.
> >
> > As mentioned above, I agree with the one-off cost of the SVD, although I don't think 20 000 variables can really be considered large scale. With $n$ variables, even if $A$ is sparse, its SVD will require $n^2$ storage, which quickly limit how high we can go.
> >
> > > Response: Yes, this can be done. For example one could use {\tt DYS-net} with regret as a loss function:
> >
> > In line with the recommendation of reviewer UhYG, I think framing this as a generic differentiable QP layer would make a lot of sense, instead of focusing on imitation learning.

---

> > > ### Author Response · Authors · 2024-07-02
> > > **Regarding scale and framing**
> > >
> > > 1. _although I don't think 20 000 variables can really be considered large scale_
> > >
> > > **Response:** Fair enough, but perhaps this is a matter of perspective. What we are trying to convey in this paper is that `DYS-net` works well for problems significantly larger than what has previously been studied in the literature. Extending decision-focused learning to truly large scale problems is an interesting yet difficult problem, we only claim to make a step towards this.
> > >
> > > 2. _In line with the recommendation of reviewer UhYG, I think framing this as a generic differentiable QP layer_
> > >
> > > **Response:** In our response to reviewer UhYG we outline how our results pertain only to QPs with objective functions of the form $w^{\top}x + \gamma |x|^2$. So, we canot (yet) claim `DYS-net` works for generic QPs.
> > >
> > > In other responses to reviewer UhYG we explain our rationale for focusing on the $\ell_2$ loss, as opposed to a generic task loss.
> > >
> > > Hopefully this addresses this point satisfactorily?

---

> > > > ### Comment · Reviewer_DjLc · 2024-07-03
> > > >
> > > > Thank you, I'm satisfied with these answers, and with your explanation to Reviewer UhYG of how the forward and backward pass are intertwined.

---

> ### Author Response · Authors · 2024-06-21
> **Response to Reviewer DjLc (part 3)**
>
> 7. _In the proof of Theorem 6, can you give a reference for the result on the first principal angle between subspaces with zero intersection? It was not obvious to me intuitively, although I don't doubt it_
>
> **Response:** We’re not aware of a good reference for this, but are happy to provide a quick proof. From the variational definition of principle angle $\tau$ given in equation (45):
>
> $$
>     \tau = 0 \Leftrightarrow \cos(\tau) = 1 \Leftrightarrow \langle u, v\rangle = 1.
> $$
> where $u \in \mathcal{H}_1^{\bot}$ and $v \in \mathcal{H}_{2,z}^{\bot}$, both with unit norm. By Cauchy-Schwarz,
>
> $$
>     \langle u, v \rangle \leq \|u\|\|v\| = 1,
> $$
>
> with equality holding if and only if $u=v$. But then $u \in \mathcal{H}_1^{\bot} \cap \mathcal{H}_{2,z}^{\bot}$ and is non-zero, i.e., the intersection is non-trivial.
>
> 8. _You claim that you cannot use PyEPO for grid sizes larger than 30 because it automatically calls Gurobi. However, you can actually plug any algorithm into PyEPO with a user-defined model, so a Dijkstra implementation is also possible instead of Gurobi._
>
> **Response:** Ah, we did not know this; thanks for pointing it out!
>
> 9. _I'm very surprised that CVX is unable to load when the grid size exceeds 30. Is the constraint matrix $A$ encoded in a sparse manner?_
>
> **Response:** Note that we are using the {\tt cvxpylayers} package,as provided on their github repo, which calls {\tt CVX}. The number of variables in the shortest path problem is roughly the number of edges squared. For $50\times 50$ grid, this yields $4900$ variables. This is the bottleneck in using {\tt cvxpylayers}. Even for smaller grids, eg 30-by-30, {\tt cvxpylayers} throws the warning {\tt UserWarning: Your problem has too many parameters for efficient DPP compilation.}  We did some digging through the source code, its possible that by substantially refactoring the {\tt cvxpylayers} code base one could gain some computational advantage on the forward pass by using sparse $A$. However, it's not clear this will improve the efficiency of the backward pass, and is beyond the scope of this work.
>
> 10. _Even if the cost vector given by $w(d)$ is nonnegative to begin with, the added random vector of PertOpt or the black box interpolation of BB might change so much that one of its components becomes negative. This is the reason behind the multiplicative perturbation introduced in [Dalle et al., 2022], which preserves the sign. Of course such a behavior is rarely observed if the perturbation is small enough, and the original code from Berthet et al implicitly handles it with (what I think is) an incorrect Dijkstra implementation. I'm not asking you to change your experiments because of that, but I would be curious to see what you think, and if it is worthy of discussion._
>
> **Response:** Thanks for the reference! Yeah the code of Berthet et al is sort of a “non-backtracking” Dijkstra, as it keeps a list of vertices that have been visited and does not visit them again. This somewhat solves the issue of negative edge weights as it prevents loops.
>
> The idea of a multiplicative perturbation instead of an additive perturbation is a good one; we have added some discussion about this to our paper.
>
> 11. _Typos_
>
> **Response:** Fixed. Thank you!

---

> > ### Comment · Reviewer_DjLc · 2024-07-01
> >
> > > Response: Note that we are using the {\tt cvxpylayers} package,as provided on their github repo, which calls {\tt CVX}. The number of variables in the shortest path problem is roughly the number of edges squared. For 50x50 grid, this yields 4900 variables.
> >
> > Can you give more details on this? The formulation I have in mind for the shortest path problem uses just one variable per edge. Perhaps you meant the number of vertices squared? In which case it is a very bad overestimate for the number of edges.

---

> > > ### Author Response · Authors · 2024-07-01
> > > **Correction to the number of variables**
> > >
> > > Apologies! We explained this terribly. What we meant to say was for a $k\times k$ grid the number of variables, which is equal to the number of edges, is $\mathcal{O}(k^2)$. So, for a $50\times 50$ grid the number of variables is a small multiple of $50^2 = 2500$.

---

### Review · Reviewer_4Zt6 · 2024-05-15

**Summary Of Contributions:**

The problem considered in this paper is contextual optimization, where a parameter of an optimization problem is unknown but depends on the context. The goal is to learn a neural network to map from the context to the parameter, minimizing the final discrepancy of the solution to the optimization problem. As this requires embedding the optimization problem into the neural network training, an implementation of a differentiable optimization layer is necessary. Moreover, it is assumed that in the training data, the true parameter vector is not recorded; only the solution and the context are present in the training data. While this problem has been covered in many recent works, as noted in Section3, the focus of this paper is on efficiency.

Implementing a projection gradient descent for this problem is difficult, as the projection must not only ensure $Ax=b$ but also $x \geq 0$. Put simply, finding even a *feasible* solution for a combinatorial optimization problem is inherently challenging. The projections $P_{C_1}$ and $P_{C_2}$ are associated with satisfying the constraints $Ax=b$ and $x \geq 0$, respectively. While combining these projections, $P_{C_1} \cap P_{C_2}$, presents difficulties, implementing each projection individually, either $P_{C_1}$ or $P_{C_2}$, is relatively straightforward. Theorem 2 addresses this by posing the projection as a fixed-point iteration, where $P_{C_1}$ and $P_{C_2}$ appear separately. The proposed technique is supported by experimental evaluation.

**Audience:**

Yes

**Broader Impact Concerns:**

I do not see any immediate requirement of mentioning broader impact concerns.

**Claims And Evidence:**

Yes

**Requested Changes:**

- How we can come from Equation 19 from Equation 18? [critical]
- Elaborate the explanation of the result (5.1.3 contains only one sentence) [strengthen]
- Consistency between Figure 2 and Figure 4 [crucial]

**Strengths And Weaknesses:**

- Strengths:



1. The paper presents its claims with proof.



2. The paper proposes a novel descent direction for differentiating combinatorial optimization problems.



- Weakness:



The experimental evaluation is relatively weak. Especially, in the warcraft dataset, it performs significantly worse than BB-net.



- Questions:

1. In Algorithm 1 Line 3, should it not be $P_{C_2} (Z^K)\approx x_{\Theta}$? Because in Equation 17 , $x_{\Theta} = P_{C_2} (Z_\Theta)$.

Similarly, it's not clear to me how  $P_{C_1}$ appear in Eq. (19), to me it seems it would be   $P_{C_2}$.

2. Can you comment on how to set the value of hyperparameter $\alpha$ and $K$. Obviously they could be selected by hyperparameter tuning; but my question is what would be a suitable range to look for?

3. Both Figure 2 (a) (b) and Figure 4 are for the shortest path problem. However, there are some discrepancies while comparing them. For example, the regret vs grid size plot for BB in Fig 2 (a) does not match with Fig 4 (c) till Grid size 30. In Figure 2 (a)regret is increasing with grid size; which is not the case in Figure 4 (c).

4. Training time of PertOpt is sometime higher than BB, sometime lower. Why would this happen? How many Monte Carlo samples have been used for PertOpt?

5. Do you have any intuition why BB performs better than DYS  by such a significant margin in the Warcraft problem? I am not sure what do you mean by "discrete" nature of cost vector?

6. Finally, in my opinion, DYS should be compared with CVX, as both of them are implementation of differentiable optimization layers, which do not require another solver (Gurobi for example). BB and PertOpt, on the other hand, improve finite difference approximation and require an optimization solver. Can you replace the Warcraft experiment with another problem, where you can run CVX?

---

> ### Author Response · Authors · 2024-06-21
> **Response to Reviewer 4Zt6**
>
> 1. _In Algorithm 1 Line 3, should it not be_ $P_{C_2} (z^K)\approx x_{\Theta}$ ...
>
>   **Response:** Yes, you are correct. That was a typo on our end; we have fixed it in the revision, and thank you for catching this!
>
>
>  2. _Can you comment on how to set the value of hyperparameter $\alpha$ and $K$. Obviously they could be selected by hyperparameter tuning; but my question is what would be a suitable range to look for?_
>
>  **Response:**  In all our experiments with DYS-net we used $\alpha = 0.05$ and $K = 1,000$, with an early stopping condition of $\|z_{k+1} - z_k\|_2 \leq {\text\tt tol}$ with ${\text\tt tol} = 0.01$. We have added a remark regarding this to our paper.
>
>  3. _Both Figure 2 (a) (b) and Figure 4 are for the shortest path problem. However, there are some discrepancies while comparing them. For example, the regret vs grid size plot for BB in Fig 2 (a) does not match with Fig 4 (c) till Grid size 30. In Figure 2 (a)regret is increasing with grid size; which is not the case in Figure 4 (c)._
>
>  **Response:** These two experiments use different implementations of BB (Fig 2(a) uses the BB implementation bundled in PyEPO, while Fig 4 uses the original implementation provided in the package blackbox-backpropagation (Vlastelica et al 2019). Importantly, the implementation using blackbox-backpropagation uses Dijkstra's algorithm while that in PyEPO uses the LP formulation and a call to Gurobi. Finally, we use different values of $\lambda$ in the experiments ($\lambda=5$ for PyEPO, $\lambda=100$ for blackbox-backpropagation) as we found by hand-tuning this parameter we could squeeze out slightly better performance.
>
>  4. _Training time of PertOpt is sometime higher than BB, sometime lower. Why would this happen? How many Monte Carlo samples have been used for PertOpt?_
>
>  **Response:** We use 3 samples for PertOpt in all experiments except the warcraft one. The greatest variability in train time is for the (NP) knapsack problem. Empirically, we observed that the run-time Gurobi applied to this problem varied quite a bit for different cost vectors $w(d)$.
>
>  5. _Do you have any intuition why BB performs better than DYS by such a significant margin in the Warcraft problem? I am not sure what do you mean by "discrete" nature of cost vector?_
>
>  **Response:** In the Warcraft problem, the cost to traverse a cell can only take on one of four values (see Figure 5, right). For the other shortest path problem, the cost to traverse a cell can take on infinitely many different values. Thus the cost vector for Warcraft is discrete, while the cost vector for shortest path is continuous.
>
>  We have redone this experiment, and `DYS-net` now does substantially better, see "General Response".
>
>  6. _Finally, in my opinion, DYS should be compared with CVX, as both of them are implementation of differentiable optimization layers, which do not require another solver (Gurobi for example). BB and PertOpt, on the other hand, improve finite difference approximation and require an optimization solver. Can you replace the Warcraft experiment with another problem, where you can run CVX?_
>
>  **Response:** We have implemented CVX for the Warcraft problem, see "General Response".
>
>  7. _How we can come from Equation 19 from Equation 18? [critical]_
>
>   **Response:** Following the works of (Fung et al, 2022), the formula for $p_{\Theta}$ in equation 19 arises from a zeroth order Neumann expansion of $J_\Theta$. We have included a brief note about this in the paper.
>
>
>   8. _Elaborate the explanation of the result (5.1.3 contains only one sentence) [strengthen]_
>
>   **Response:** We have done so.
>
>   9. _Consistency between Figure 2 and Figure 4 [crucial]_
>
>   **Response:** As we hoped to convey above, these two experiments used different implementations of the same algorithm, different base solvers (Dijkstra's algorithm vs LP formulation + Gurobi) and different hyperparameter values.

---

> > ### Comment · Reviewer_4Zt6 · 2024-06-22
> > **Response to Authors' Comment**
> >
> > Thank you very much for your reply and for updating the paper with the new experimental results and corrections. Currently, I see that the experimental evaluation indicates that CVX and DYS outperform BB and PertOpt. Among CVX and DYS, CVX delivers better decision quality, while DYS is significantly faster due to its use of successive projections as matrix operations in the forward pass.
> >
> > Thank you for clarifying the differences between Figures 2 and 4. However, I still believe this might confuse readers. It may be beneficial to use the same hyperparameters in both cases, as stated in the paper: `We tuned the hyperparameters for each approach to the best of our ability on the smallest problem (5-by-5 grid) and then used these hyperparameter values for all other graph sizes.`
> >
> > Regarding the Warcraft experiment, it is true that "the cost can only take on one of four values," but the true costs are not used in training, so I do not see how does this make an impact. Regardless, after revising the experiments, DYS has a lower regret than BB, so you might want to remove or rephrase that paragraph.
> > Also, it seems a bit unusual to assume that the true parameter $w$ is not available in the training data but is present in the test data. However, I understand this is the premise the paper is set on.
> >
> > On the discussion with UhYG about approximating LPs/ILPs using QPs, I wonder if DYS requires QP approximation. In Eq (16), setting $\gamma$ to $0$ doesn't seem to immediately present any numerical issues. If this is feasible, you might be able to claim "differentiating through LPs" without needing QP approximation. I haven't verified whether setting $\gamma$ to $0$ would cause problems, so I am interested in your thoughts on this.

---

> ### Author Response · Authors · 2024-07-02
> **Response to Reviewers comments**
>
> 1. _Regarding the differences between Figures 2 and 4_
>
> **Response:** Respectfully, we are hesitant to rerun the large-scale shortest path experiments as they took several weeks of GPU time. We are also hesitant to change the hyperparameters in the `PyEPO` experiments, as these are selected by hyperparameter tuning done by the authors of the `PyEPO` package. So, we will make it clearer in our paper that different sets of hyperparameters are used, and that this resulted from a good-faith attempt to compare `DYS-net` to the strongest versions of benchmark algorithms that we could implement.
>
> 2. _Regarding the Warcraft experiment_
>
> **Response:** Because the entries of the true cost vector in the Warcraft experiment take on only four, well-separated values (specifically: 0.8, 1.2, 7.7, and 9.2) an algorithm that learns a rough approximation to the true cost vector will likely produce the correct shortest path. In the `PyEPO` shortest path experiment, this is not the case. We think this impacts the training dynamics, even if the true parameter $w$ is not available, and explains why `BB-net` and `PertOpt-net` perform poorly on the `PyEPO` experiments. But, this is speculative, so we will rephrase this discussion.
>
> 3. _Regarding the discussion with UhYG about approximating LPs/ILPs using QPs, I wonder if DYS requires QP approximation..._
>
> **Response:** We did try setting $\gamma = 0$ experimentally, and this totally failed --- `DYS-net` did not improve with training at all. Note that $\gamma > 0$ is crucial in the proofs of Theorem 2 and Corollary 7.

---

> > ### Comment · Reviewer_4Zt6 · 2024-07-03
> > **Response to Authors' Comment**
> >
> > Thank you for the reply. I agree with your responses.
> > I also agree with reviewer UhYG that it should be made clear in the paper that `DYS-Net` is applicable when the true parameter is present in the training data and the model is trained to minimize regret. However, in such scenarios, while `DYS-Net` is efficient, SPO+ might outperform DYS-Net in terms of solution quality.

---

### Review · Reviewer_UhYG · 2024-06-03

**Summary Of Contributions:**

Summary of the Paper:
The paper proposes a method for ‘learning to solve’ mixed-integer linear programs, with uncertain coefficients. It is based on a  convex relaxation of the MILP problem’s feasible set, combined with quadratic regularization to allow differentiation through the approximated problem. The paper claims competitive accuracy with other methods on its test problems, with a decided advantage in training efficiency.

Summary Review:
The paper, to the best of my evaluation, is technically solid and sound. Its claims seem well-justified, though I did not check the proofs. The proposed method uses an interesting mix of very modern techniques to try and minimize computational effort in solving a difficult problem. On the other hand, it suffers from a lack of clarity in its key insights and contributions, so that I am ultimately unconvinced of its overall significance, despite being interesting. The “Strengths” and “Weaknesses” below elaborate on these points.

**Audience:**

Yes

**Broader Impact Concerns:**

I don't foresee any significant ethical concerns.

**Claims And Evidence:**

No

**Requested Changes:**

Many of my criticisms (the first four Weaknesses bullets) would be addressed if the authors were willing to reframe their paper as one which learns the coefficients of QP problems rather than ILPs. This would require a change of title and significant reframing of the intro sections. Depending on the opinion of other reviewers, I may see this as a major issue with the paper, because it means the paper is not supporting its own claims. I also welcome the authors to give their contrasting view on the issue.

The authors should also better motivate their use of Davis-Yin splitting in the forward pass (bullet 5 in Weaknesses), and clarify where their computational advantage actually stems from (forward or backward pass).

**Strengths And Weaknesses:**

Strengths:
- The paper is well-written with good organization, related work and technical descriptions that make the paper easy to read. Overall a high-quality presentation.
- The paper convinces that its approach is sound and well-justified. It  makes use of several very recent methodologies as part of its overall approach. I enjoyed reading it and did not have any major objections on a technical level.
- The technical development and design choices make sense and are sufficiently explained, and the theory is thoroughly described, though I may have overlooked some details. The proofs, based on the gradients' validity as a descent direction, are a nice inclusion.
- The experimental evaluations are clear, well-illustrated and decently thorough, on the test problems that were chosen.


Weaknesses:
- The title is misleading, implying that the paper will be about “Learning to Solve ILP problems”. Ultimately, the proposed method is evaluated on “Predict-Then-Optimize (PtO)” tasks against methods in that domain. The paper makes no contribution in learning to solve ILP problems, but rather in learning the uncertain coefficients which may appear in a ILP problem. In this paper's framework, the ILP problem must be solved (at test time) by conventional methods, and not a learned model, unless I am mistaken.

- Further, the Related Work is framed in a misleading way, which contributes to this paper's confusing framing of its own contributions. First, I think it's generally accepted that "Learning to Optimize (L2O)" is about speeding up the solution of optimization problems. This paper expands the notion of L2O to include learning the uncertain coefficients of an optimization problem, but this setting already has a name (Predict-Then-Opimize, or Decision-Focused Learning). Moreover, some works cited in the subsection "Differentiable Combinatorial Optimizers" are miscategorized. For example, Berthet 2020 is explicitly only developed for linear programs with continuous variables, and not ILPs. Elmachtoub 2022 does not give a differentiable optimizer, but rather only a surrogate subgradient for its resulting regret loss.

- Also in terms of its framing of technical contributions: not only does the paper not show how to learn to solve ILPs (as implied by the title, see the first bullet), it also does not really even show how to learn the coefficients of ILPs. It only shows how to learn the coefficients of convex continuous Quadratic Programming problems, which is a much lesser feat. It's true that a couple other works have claimed to learn ILP coefficients in Predict-Then-Optimize, when what they really do is form a QP approximation to the ILP and learn its coefficients instead. So there is precedent for what the authors do here. However, I think it's an unjustified practice, since it assumes that QP relaxations of ILPs are valid in general, and this has never been sufficiently demonstrated in the literature (unless I'm mistaken). This may seem like a nitpick, but I think it's important, because we are now seeing it become "folklore" that ILPs can simply be replaced with QPs in the general case. Unless this can be justified, it seems like a misleading trend in the literature motivated by "up-selling" of the papers' contributions, where any QP method can be sold as an ILP method. I'm open to push-back from other reviewers on this view.

- Because the paper claims to handle ILP problems, the experimental evaluation is arguably weak. We are only evaluating the method on two problem types, knapsack and shortest path. Of those, shortest path is not properly a ILP problem, it is a continuous LP with totally unimodular constraints which guarantee integral solutions. But it is solved as an LP, or even n log n methods like Dijstra's. So the shortest path experiment does not demonstrate the premise of the paper, which promises to learn to solve ILPs. The knapsack problem has been shown in empirically in other works to admit a good QP relaxation for Predict-Then-Optimize, so it's not surprising to see as the other experiment. For the paper to justify its claims, it needs to evaluate on another NP-hard, proper ILP problem (in my opinion).

- About the choice of forward pass. The QP approximation is solved by Davis-Yin splitting. I think the paper could do a better job of motivating this. Given that the paper is about solving large-scale problems, this cannot be the fastest way to solve the QP in general. In fact, we seem to have the option to use any black-box optimization solver to replace this forward pass, and proceed directly to the backward pass in Step 4. Is this correct? If so, it should be made clear. This is important to question, since the forward pass is part of an overall method which claims efficiency advantages over other methods. The results of Figure 2 do not indicate whether the efficiency gains stem from the forward pass or the backward pass.

- This DYS-net method requires pre-computation of optimal solutions (x*) to each instance of the NP-Hard MILP problems that make up the training set. They are needed for its loss function. Importantly, none of the evaluated baseline methods require this step. If this is correct, I think it presents a major challenge to the claimed efficiency gains over other methods, which use the ILP's empirical objective value as their loss function.

- Beyond the efficiency drawback of requiring precomputed optimal solutions, I expect that this requirement will also lead to issues in training. Since the MILP problems are NP-Hard, we can not expect to find their true optimal solutions in general. If we settle for sub-optimal precomputed solutions, you cannot expect the training set to preserve the true relationship between unknown problem parameters and the optimal solutions. For this reason and the one above, the loss function seems flawed.

- Building on the above three points, I would conjecture that the reason for using the regression loss (rather than minimizing the objective as a loss) is because the Davis-Yin forward pass will tend not to converge in finite iterations in the latter case. However, replacing the Davis-Yin method with a black-box forward pass optimization would seem to resolve this issue, guaranteeing feasibility so that we can use f as a loss function and avoid the issue of precomputed optimal solutions.  Is this correct, and if so, why not take this type of approach?

-It's not clear what is the significance of the theoretical results. Are they special cases of previously established results, or is this theory novel? This is a question, not necessarily a weakness.

---

> ### Author Response · Authors · 2024-06-21
> **Response to Reviewer UhYG (part 1)**
>
> We thank the reviewer for their time and for their constructive feedback! We appreciate your description of our paper as "well-written" and a "high quality presentation". Before addressing your remarks point by point, we have two clarifying high-level responses to make.
>
> **Training Data:** There are two closely related learning problems in the literature, depending on whether the training data is (i) (context, cost vector); in our notation $(d, w(d))$, or (ii) (context, solution); in our notation $(d, x^{\star}(d))$.
> Our focus is exclusively on the latter. Learning from type (ii) data is strictly harder than learning from type (i) data, as $x^{\star}(d)$ can in principle be computed from $w(d)$ but not _vice versa_. As elaborated on in point 6 below, in some settings it is reasonable to assume (nearly) optimal solutions/decisions $x^{\star}(d)$ can be observed.
>
> **Title, Related Work, and Framing of contributions:** We thank the reviewer for their critical yet constructive feedback on this point. We agree with some of your comments, and have modified our paper accordingly. In particular, we propose changing the title of this work to _Modeling Integer Linear Programs with Quadratic Regularization and Davis-Yin Splitting_. This acknowledges that, as you point out, we do not propose a new method for solving ILPs, but rather propose an efficient way to construct a useful model of an ILP with unknown parameters. On other points we shall explain our reasoning more precisely and welcome further discussion.

---

> > ### Author Response · Authors · 2024-06-21
> > **Response to Reviewer UhYG (part 3)**
> >
> > 4. _Because the paper claims to handle ILP problems, the experimental evaluation is arguably weak. We are only evaluating the method on two problem types, knapsack and shortest path. Of those, shortest path is not properly a ILP problem, it is a continuous LP with totally unimodular constraints which guarantee integral solutions. But it is solved as an LP, or even n log n methods like Dijstra's. So the shortest path experiment does not demonstrate the premise of the paper, which promises to learn to solve ILPs. The knapsack problem has been shown empirically in other works to admit a good QP relaxation for Predict-Then-Optimize, so it's not surprising to see as the other experiment. For the paper to justify its claims, it needs to evaluate on another NP-hard, proper ILP problem (in my opinion)._
> >
> >   **Response:** To push back a little, although we focus on two problem types we do four distinct experiments. It *is* surprising this QP relaxation approach works for the knapsack problem because, as you point out above, this cannot be deduced from theory. Also, at the time of writing, there were relatively few datasets in the P-then-O/DFL literature with modifiable problem dimension.
> >
> >   Nonetheless, if the reviewer deems it essential we are happy to conduct some additional experiments. Setting them up, tuning hyperparameters etc may, however, require a bit of time.
> >
> >   5. _About the choice of forward pass. The QP approximation is solved by Davis-Yin splitting. I think the paper could do a better job of motivating this. Given that the paper is about solving large-scale problems, this cannot be the fastest way to solve the QP in general. In fact, we seem to have the option to use any black-box optimization solver to replace this forward pass, and proceed directly to the backward pass in Step 4. Is this correct? If so, it should be made clear. This is important to question, since the forward pass is part of an overall method which claims efficiency advantages over other methods. The results of Figure 2 do not indicate whether the efficiency gains stem from the forward pass or the backward pass._
> >
> >   **Response:** We focus on first-order methods (which include operator-splitting schemes) as they tend to perform better, in terms of wall-clock time, for large problems where high-accuracy solutions are not essential (think (stochastic) gradient descent vs Newton's method for training neural networks).
> >
> >   Projected gradient descent performs poorly here for general $\mathcal{C}$ because the projection onto $\mathcal{C}$, which is computationally expensive, needs to be computed _at every iteration of every forward pass for every sample of every epoch_.
> >
> >   Davis-Yin splitting, because it is a three operator splitting technique, is able to decouple the polytope constraints into relatively simple constraints. This is shown in detail in (McKenzie et al 2021), and used in many other places e.g. Section 4.1 of (Pedregosa & Gidel 2018). Thus projection onto $\mathcal{C}$ can be avoided; only projections onto $\mathcal{C}1$ and $\mathcal{C}2$ are needed.
> >
> >   The avoidance of this costly projection gives DYS an advantage in terms of wall-clock time for both the forward pass and the backward pass. See Appendix F of (Mckenzie et al 2021) for an experiment illustrating this and further discussion.
> >
> >   Thus, we believe that DYS is one of the fastest methods for this particular problem setting, with efficiency gains from both the forward and backward passes. We have expanded the discussion of the forward pass in Section 4 to make this clearer.
> >
> >   Yes, in theory one can mix-and-match the methods used for the forward and backward pass; see (Bolte et al, 2024) for an interesting discussion of this approach. However, in practice this makes implementation in code much more involved. Using the same method in the forward and backward pass allows for easy implementation within common deep learning packages (e.g. PyTorch), which in our opinion is strong motivation for doing so.

---

> ### Author Response · Authors · 2024-06-21
> **Response to Reviewer UhYG (part 2)**
>
> --------
>
> ### Point-by-point responses
>
> 0. _The title is misleading, implying that the paper will be about “Learning to Solve ILP problems”. Ultimately, the proposed method is evaluated on “Predict-Then-Optimize (PtO)” tasks against methods in that domain. The paper makes no contribution in learning to solve ILP problems, but rather in learning the uncertain coefficients which may appear in a ILP problem. In this paper's framework, the ILP problem must be solved (at test time) by conventional methods, and not a learned model, unless I am mistaken_
>
> **Response:** We have changed the title of our paper to avoid this confusion. Our framework allows, but does not require, the use of conventional methods to solve the ILP at test time (see the "Implementation" paragraph in Section 4).
>
> 1. _Further, the Related Work is framed in a misleading way, which contributes to this paper's confusing framing of its own contributions. First, I think it's generally accepted that "Learning to Optimize (L2O)" is about speeding up the solution of optimization problems. This paper expands the notion of L2O to include learning the uncertain coefficients of an optimization problem, but this setting already has a name (Predict-Then-Opimize, or Decision-Focused Learning)_
>
> **Response:** The bulk of the predict-then-optimize (P-then-O) literature focuses on training data of type (i), as described above. This is decidedly not the focus of our work, and so we deliberately avoided the term "predict-then-optimize" to prevent confusion. In the inverse problem community, learning-to-optimize (L2O) can refer to the problem of learning a regularizer (Gilton et al 2021), which is akin to our setting in which we learn the objective function. Here the emphasis is on better, not faster, image reconstruction. This is also noted in the L2O survey (Chen et al 2022): "The learned optimizer may also return a higher-quality solution to a difficult task than classic methods, given a similar amount of computing budget.''  Moreover, the core problem in using implicit neural networks within L2O is differentiating through a solution or fixed point $x^{\star}_{\Theta}$, see e.g. (Gilton et al 2021). As our technical contributions address this issue, we think it is productive to make this connection.
>
> We have reworked and reworded the Related Work section to make it clearer that, while our work is related to L2O, it is better situated within the decision-focused learning literature
>
>  2. *Moreover, some works cited in the subsection "Differentiable Combinatorial Optimizers" are miscategorized. For example...*
>
>  **Response:** We have changed the paragraph heading to **Decision-focused learning for LPs** to avoid any confusion.
>
>  3. _Also in terms of its framing of technical contributions: not only does the paper not show how to learn to solve ILPs ...It only shows how to learn the coefficients of convex continuous Quadratic Programming problems, which is a much lesser feat. It's true that a couple other works have claimed to learn ILP coefficients in Predict-Then-Optimize, when what they really do is form a QP approximation to the ILP and learn its coefficients instead. So there is precedent for what the authors do here. However, I think it's an unjustified practice, since it assumes that QP relaxations of ILPs are valid in general, and this has never been sufficiently demonstrated in the literature..._
>
>  **Response:** You raise a very important point.  We would suggest that the justification lies in the results. Using the QP relaxation as a proxy for the true ILP works well, at least in some cases. This is demonstrated empirically in our work, and in the many other P-and-O/DFO/L2O works we build upon. But empirical success is not an excuse for epistemic hubris. We have changed the word "Learning" in the title to "Modeling", and add some discussion on the potential pitfalls of using a QP as a proxy for an ILP to Section 2.

---

> > ### Comment · Reviewer_UhYG · 2024-06-21
> >
> > Thanks making efforts to accomodate my feedback and detailing your views on the matter of terminology. I agree with the authors' proposed changes; they address some of my initial concerns.
> >
> >
> > With regards to that "... the justification lies in the results...", I still disagree. It's not sufficient that QP relaxation to ILP works "at least in some cases". The principle needs to be demonstrated on more than just knapsack problems. I won't recommend rejection on those grounds, but I strongly recommend that care is taken not to further promote the notion that such approach is valid on general ILP's, which to my knowledge is still unsubstantiated despite having taken hold in the literature.

---

> > > ### Author Response · Authors · 2024-07-02
> > > **Validity of QP relaxation to ILPs**
> > >
> > > Noted. We will edit the discussion of this in our paper to make it clear that we can at most say the QP relaxation approach works for the knapsack problem, but that this approach might not work for ILPs with large integrality gap. We will also keep this in mind when communicating the results of this paper.

---

> ### Author Response · Authors · 2024-06-21
> **Response to Reviewer UhYG (part 4)**
>
> 6. _This DYS-net method requires pre-computation of optimal solutions (x*) to each instance of the NP-Hard MILP problems that make up the training set. They are needed for its loss function. Importantly, none of the evaluated baseline methods require this step. If this is correct, I think it presents a major challenge to the claimed efficiency gains over other methods, which use the ILP's empirical objective value as their loss function._
>
>  **Response:** This is not true, all baseline methods we considered require the solutions $x^\star$ for training (i.e. they using training data of type (ii)). In applications we are envisaging, these $x^{\star}$ can be observed. For example, in shortest path problems, the routes taken by experienced taxi drivers are nearly optimal. We only use $w(d)$ for model evaluation. Methods which use $w(d)$ during training, i.e. training data of type (i), are solving a different problem (usually called predict-then-optimize). Note that having access to $w(d)$ at training time is a strictly easier problem than having access only to $x^{\star}(d)$ --- given $w(d)$ one can in principle find $x^{\star}(d)$ but not vice versa --- so these two classes of methods cannot be compared in an apples-to-apples way.
>
>
>  7. _Beyond the efficiency drawback of requiring precomputed optimal solutions, I expect that this requirement will also lead to issues in training. Since the MILP problems are NP-Hard, we can not expect to find their true optimal solutions in general. If we settle for sub-optimal precomputed solutions, you cannot expect the training set to preserve the true relationship between unknown problem parameters and the optimal solutions. For this reason and the one above, the loss function seems flawed._
>
>  **Response:** End-to-end learning of the unknown problem parameters given sub-optimal solutions is an interesting topic of active research, see [1]. We reiterate that for the applications in which we are interested, the training data is of the form $(d,x^{\star}(d))$, i.e. (potentially noisy) optimal solutions are observed, not computed.
>
>  8. _Building on the above three points, I would conjecture that the reason for using the regression loss (rather than minimizing the objective as a loss) is because the Davis-Yin forward pass will tend not to converge in finite iterations in the latter case. However, replacing the Davis-Yin method with a black-box forward pass optimization would seem to resolve this issue, guaranteeing feasibility so that we can use f as a loss function and avoid the issue of precomputed optimal solutions. Is this correct, and if so, why not take this type of approach?_
>
>  **Response:** Our focus on the regression loss $\|x^{\star}(d) - x_{\Theta}(d)\|^2$ was inspired by our reading of works like (Vlastelica et al 2019) and (Berthet et al 2020) which focus on training data of type (ii). The convergence of DYS is independent of the loss function; it only depends on how you compute $x_{\Theta}(d)$, not on how you measure the "goodness" of $x_{\Theta}(d)$. Apologies if we have not understood your question correctly.
>
>  We did do some preliminary experiments on training data of type (i). We used the objective function $w(d)^{\top}x_{\Theta}(d)$ as the training loss but computed $x_{\Theta}(d)$ in exactly the same way as for the regression loss experiments. We observed that DYS was outperformed by SPO+ (Elmachtoub & Grigas, 2022) in this setting, and so we did not pursue this further.

---

> > ### Comment · Reviewer_UhYG · 2024-06-21
> >
> > In Response to "This is not true, all baseline methods we considered require the solutions..."
> > - Those baseline methods are based on generic differentiable layers, and do not assume a target data setting which requires precomputed solutions; for a thorough demonstration of their performance on type (i) data please see Mandi et. al. "Decision-Focused Learning: Foundations, State of the Art, Benchmark and Future Opportunities".
> >
> > - I believe a short discussion of your method's applicability to type (i) data is all that's needed to address this issue in the paper, no evaluation required.

---

> > > ### Author Response · Authors · 2024-07-02
> > > **Regarding baseline methods**
> > >
> > > In response to "Those baseline methods are based on generic differentiable layers..."
> > >
> > > - You're absolutely correct, thanks for pointing this out!

---

> ### Author Response · Authors · 2024-06-21
> **Response to Reviewer UhYG (part 5)**
>
> #### Requested Changes
>
>  1. _Many of my criticisms (the first four Weaknesses bullets) would be addressed if the authors were willing to reframe their paper as one which learns the coefficients of QP problems rather than ILPs. This would require a change of title and significant reframing of the intro sections. Depending on the opinion of other reviewers, I may see this as a major issue with the paper, because it means the paper is not supporting its own claims. I also welcome the authors to give their contrasting view on the issue._
>
>  **Response:** To summarize the various responses above, we will modify the title ("Learning" to "Modeling") and add additional context to our literature review. We acknowledge that what the DYS-net methodology is doing is building a good model of the underlying ILP, not solving it.
>
>  We do not think a reframing focused on QPs would be productive. Note that all QPs considered in this work have objective functions of a very particular form: $w^{\top}x + \gamma \|x\|^2$. All our theoretical results of Section 4 pertain to QPs of this form, not general QPs. Analyzing just this case is technically demanding, as the results of Section 4 (and their proofs) show. Analyzing this case is also motivated by pressing real-world applications; namely that relaxed ILPs that we consider. Studying the full QP case would be an interesting topic for future work and, as a side note, would link to interesting applications such as the (contextual) quadratic knapsack problem that would build upon the results of our current paper.
>
>  2. _The authors should also better motivate their use of Davis-Yin splitting in the forward pass (bullet 5 in Weaknesses), and clarify where their computational advantage actually stems from (forward or backward pass)._
>
>  **Response:** We will expand the discussion of the forward pass in Section 4 to motivate the use of DYS better. The computational gains stem from both the forward and backward pass
>
> ------
>
> #### References
>
> [1] _DataSP: A Differential All-to-All Shortest Path Algorithm for
> Learning Costs and Predicting Paths with Context_ Alan A. Lahoud, Erik Schaffernicht, Johannes A. Stork, _arxiv preprint_ (2024).

---

> > ### Comment · Reviewer_UhYG · 2024-06-21
> >
> > In reponse to "We do not think a reframing focused on QPs would be productive. Note that all QPs considered in this work have objective functions of a very particular form..."
> >
> > - Thank you for correcting my mistaken view on that point. I understand the distinction. This seems easily resolved by more properly framing the method as one which works for LPs, not QP's.

---

> ### Comment · Reviewer_UhYG · 2024-06-21
>
> Responding to: "... if the reviewer deems it essential we are happy to conduct some additional experiments."
>
> - I would rather the claims were simply modified to tone down the implication that the proposed method should work on general ILPs, unless the QP relaxation principle for ILPs can be backed with references that support it with evidence on a variety of ILPs, aside from just knapsack.
>
>
>
> Responding to: "It is surprising this QP relaxation approach works for the knapsack problem because, as you point out above, this cannot be deduced from theory."
>
> - This is precisely my concern; moreso, it's not even supported empirically (not sufficiently, as far as I can tell).
>
>
>
> Responding to: "... with efficiency gains from both the forward and backward passes."
>
> - If I understand correctly, claiming efficiency gains in the forward pass (by using Daivs-Yin splitting) amounts to claiming a contribution in the classical optimization domain, in solving QPs. Even if it were true, that contribution is not attributed to this paper. Unless I'm mistaken on that point, it should be clarified in the paper that the proposed efficiency gains stem from the backward pass, not the forward pass. I think that would greatly strengthen the paper by clarifying its technical contributions.
> - If indeed the Davis-Yin approach leads to efficiency advantages in \textbf{solving} Quadratic Programs, particularly of the form considered in the paper where the quadratic term is composed of a Euclidean norm: this fact should be demonstrated in the paper. Otherwise it seems that the paper should clarify that its proposed efficiency advantage comes by virtue of its backward pass. Please let me know if my understanding is mistaken.

---

> > ### Author Response · Authors · 2024-07-02
> > **Regarding efficiency gains from both the forward and backward pass.**
> >
> > There is an entanglement between the forward and backward pass. Algorithms based on the methodology proposed by Amos & Kolter (2017), i.e. Wilder _et al_ (2019), Agrawal _et al_ (2019) (the `cvxpylayers` paper), need to solve the QP on the forward pass for both primal and dual variables; the dual variables being required for the backward pass. The aforementioned papers use some variant of the primal-dual interior point method on the forward pass, which takes a Newton-like step at each iteration. This Newton-like step involves solving a linear system of size (num. constraints + num variables) and is the computational bottleneck. On page 4 of Amos & Kolter 2017 they point out that the forward pass actually takes up the bulk of the computational time. Once both primal and dual variables at optimality are known, performing the backward pass is quick.
> >
> > Because we compute the backward pass by differentiating an operator-based optimality condition (see equation 17) instead of a KKT-based optimality condition (see the unnumbered equation immediately above equation 12) we do not need dual variables. Thus, we are free to use an operator-splitting approach for the forward pass. We use DYS, but probably other approaches are possible. This is faster, at least when the number of variables is large, than the primal-dual interior point method. It is in this sense that we gain efficiency in the forward pass. Because we use JFB, instead of solving for the exact gradient, there is an additional efficiency gain in the backward pass.
> >
> > We are not claiming any contributions to the classical optimization literature. Doubtless, a highly optimized QP solver (e.g. OSQP [1]) would solve the QP faster, although one would then need to implement a backward pass. Perhaps this would be a good topic for future work. What we are trying to convey here is that `DYS-net` is significantly faster than competing approaches to solving and differentiating through this type of QP.
> >
> > ------
> > [1] _OSQP: An Operator Splitting Solver for Quadratic Programs_ Stellato, Banjac, Goular, Bemporad and Boyd 2020.

---

> ### Comment · Reviewer_UhYG · 2024-06-21
>
> Thanks for the detailed response regarding the form of the training data. I acknowledge that supervision under precomputed solutions is a legitimate setting, and is actually distinct from "Predict-Then-Optimize".
>
> It's fair to bring up (Vlastelica et al 2019) and (Berthet et al 2020) and in light of that, I see no problem with the paper's focus on data of type (ii), at least in the experiments. I think the difference is that those papers don't limit themselves to data of type (ii) in their overall framing. They also happen to work for data of type (i).  This (DYS) paper assumes data of type (ii) from the outset.
>
> Please answer if possible: can the proposed method handle data of type (i)? If not, why not? And if so, why not frame the paper as a more generic differentiable layer, rather than assuming type (ii) data as part of its premise? In my view, this results in a lack of clarity regarding the method's capability. Especially given that the proposed method trains a prediction model of exactly those parameters that make of the type (i) target data.

---

> > ### Author Response · Authors · 2024-07-02
> > **On type (i) versus type (ii) data**
> >
> > We ran some initial experiments using `DYS-net` for type (i) data with regret as a loss function. However, we were unable to do better than the SPO+ approach, so we did not pursue this further.
> >
> > Our reading of the literature (e.g. benchmarking results of Khalil and Tang (2023) and Mandi et al (2024)) suggests that SPO+ is really hard to beat when type (i) data is available. Thus, we focus on type (ii) data as this research problem is less settled.
> >
> > We do mention, in Section 2, that losses other than the $\ell_2$ loss can be considered with only superficial changes. We could expand upon this, and frame `DYS-net` as a generic differentiable layer. However, this increased generality would, in our opinion, make this paper harder to follow. Moreover, as `DYS-net` does not appear to be superior to the SPO+ approach when type (i) data is available, the upside is not so clear to us.
> >
> > Regardless, we will add a remark on the applicability of our method to type (i) data.

---

> ### Comment · Reviewer_DjLc · 2024-07-01
>
> > Responding to: "... with efficiency gains from both the forward and backward passes."
>
> I agree with reviewer UhYG on the need to clarify respective contributions between forward and backward pass. It is also my view that the forward pass is an "implementation detail", and could be replaced with any GPU-compatible QP solver, while the backward pass is more central.

---

> > ### Author Response · Authors · 2024-07-02
> > **Regarding efficiency gains on both the forward and backward pass**
> >
> > Please see our response to reviewer UhYG on this matter.

---

> ### Comment · Reviewer_DjLc · 2024-07-01
>
> > And if so, why not frame the paper as a more generic differentiable layer, rather than assuming type (ii) data as part of its premise?
>
> After skimming the discussion, I very much agree with this suggestion.

---

> > ### Author Response · Authors · 2024-07-02
> > **Regarding framing as a generic differentiable layer**
> >
> > Please see our response to reviewer UhYG on this matter.

---

### Author Response · Authors · 2024-06-21
**General Response to Reviewers**

Thanks to all reviewers for their careful reading of our work, and for their insightful comments! We shall address their concerns point-by-point below. Here we list a few general responses.

## V2 of the paper
The pdf of our paper available on this platform has been updated in response to the reviewers comments.

## Typos etc.
A few typos occurred where the notation for the projections onto $C_1$ and $C_2$ was accidentally swapped. This has been fixed. Other typos, misspellings etc have been fixed, and we thank the reviewers for pointing these out.

## Warcraft experiment

Based on feedback from reviewer 4Zt6 we have substantially improved the warcraft experiment. Specifically:

  - We implemented a CVX based method for this problem, which performs quite well.
  - We did a modest amount of hyperparameter tuning for `DYS-net`. In particular, we found setting the strength of quadratic regularization, _i.e._ $\gamma$, to $0.5$ greatly improved the performance of {\tt DYS-net} so that it now outperforms `BB-net`. Note that for {\tt PertOpt-net} we use the hyperparameter values suggested by Berthet _et al_ (2020). For `BB-net` we tried the hyperparameter value $\lambda = 20$ as suggested by Vlastelica et al. However, we found that $\lambda =10$ performed better.
   - We found a slight error in the script we used to compute accuracy. Essentially, it was overstating the accuracy by a factor of 15. After correcting this, none of the algorithms we tested score highly in terms of accuracy. In fact, this is to be expected. When introducing this dataset, Vlastelica et al (2019) observe that the shortest path for many maps is not unique---there are many distinct paths achieving the same length. Because of this, we have removed accuracy as a metric from Table 2 and just focus on regret.
  - Finally, for each method we will run three trials and report the average. We have already done this for `DYS-net` and `CVX-net`. The experiments for `PertOpt-net` and `BB-net` are still running. We will update the draft as soon as they are complete.

## Convention on references
- In our responses, we use the same convention for references as in the paper, e.g. (Zhong \& Tang, 2021) or (Berthet et al, 2020). Additional references not in the version of the paper under review are given numerically (e.g. [4]).

---

> ### Author Response · Authors · 2024-07-05
> **Table of results for Warcraft experiment**
>
> Hi All,
>
> Following up on our previous comment, here are the complete results for the warcraft experiment:
>
> | Algorithm          | Test Normalized Regret | Time (in hours) |
> |--------------------|-------------------------|-----------------|
> | BB-net            | 0.1204              | 0.11                            |
> | PertOpt-net    | 0.1089              | 2.07                            |
> | DYS-net         | 0.0889              | 0.09                            |
> | CVX-net         | 0.0214              | 2.33                             |
>
> *Table: Results for the 12-by-12 Warcraft shortest path prediction problem. We select the model achieving the best normalized regret on the validation set. The time displayed is the time till this best normalized regret is achieved. All results are averaged over three runs.*
>
> We refactored our code, and added model selection to the training loop for BB-net and PertOpt-net. Specifically, after each epoch we check the regret on a validation set and checkpoint the model achieving lowest validation regret. The regret reported in the table is the regret on the test set for the model achieving lowest regret on the validation set.
>
> As can be seen, this model selection strategy helps a lot, particularly for `PertOpt-net`. However the analysis expressed by Reviewer 4Zt6 [here](https://openreview.net/forum?id=H8IaxrANWl&noteId=SCS2GLZDqn) still applies: "CVX and DYS outperform BB and PertOpt. Among CVX and DYS, CVX delivers better decision quality, while DYS is significantly faster..."
>
> We made two other minor changes to the experimental setup:
>  - We have dropped the "Baseline" method, as it is uninformative and does not fit neatly into our refactored training code.
>  - We report the normalized regret (defined in eq. 25 or our paper) instead of the relative regret, for consistency with the other `PyEPO` based experiments.

---

### Author Response · Authors · 2024-06-25
**Timeline for replies**

Dear reviewers,

We thank you for your timely responses! Several of us are on family vacations this week, so we will respond to your additional questions by Monday, July 1st at the latest.

All the best,
Authors

---

### Author Response · Authors · 2024-07-15
**Following Up**

Dear reviewers,

Thanks again for the time you have invested in this paper! Are there any further questions we could address?

---

> ### Comment · Action_Editor_NTsU · 2024-07-15
>
> Hi authors,
>
> We're finalizing our internal discussions at the moment. My recommendation will be sent to the Editors in Chief soon.

---

### Decision · Action_Editor_NTsU · 2024-07-16

**Recommendation:** Accept with minor revision

**Comment:**

Reviewers were unanimous in recommending acceptance of the paper, and generally feel excited about the new proposed approach for differentiating through LPs (with quadratic regularization).

There was also an internal discussion on whether we should recommend this submission for any additional certifications and/or ICLR presentations. While we are excited about the paper, the consensus is that additional recognition would require further revisions on the paper to strengthen some points. We detail our concrete comments below, and the authors can decide whether they want to make the requested revisions. If the requested changes are made satisfactorily, I will ask the Editors in Chief to add relevant certifications/recommendations, who have the ultimate authority to grant these recognitions.

Minor revisions (required for the camera ready version for acceptance to TMLR):

- Clarify with a clear acknowledgement that the focus on (features,solution) training data instead of (features,true cost parameters) is due to not being able to outperform SPO+. Please also clarify what "outperforming" means, in terms of training time and/or predictive accuracy for regret.

- Add discussion on forward vs backward pass efficiency gains. My understanding is that by using fast Jacobian-Free backpropagation for the backward pass, the forward pass no longer needs primal-dual solving (via not differentiating the KKT conditions at LP optimality), which further allows for the fast Davis-Yin solving in the forward pass. This is an interesting point, and the discussion would serve to strengthen the paper.

- Update the experimental result tables according to the rebuttal discussion.

More major revisions (required for my recommendation as AE for additional certification and/or ICLR journal-to-conference track):

- Reframe the introduction to emphasize that the main contribution is a novel generic differentiable LP layer approach, before saying that the focus on (features,solution) is due to comparison with SPO+ etc. (see above), and before mentioning that the approach is then also applied to LP relaxations of MILPs.

- Add at least some smaller experiments demonstrating the approach on (feature,true cost parameter) data, comparing with SPO+ and perhaps also Mandi and Guns (Interior Point Solving for LP-based prediction+optimisation, NeurIPS 2020), demonstrating that the proposed approach might not be fully competitive with other methods in the context of (feature,true cost) training data. Reviewers and I believe that the transparency will make the work more convincing.

- For the runtimes reported in experiments, separate out the times for the forward pass and backward pass for both cvx and DYS-net, in the context of the discussion outlined above.

- At least one more benchmark on a (mixed) *integer* linear program problem, ideally on a problem that has some decent integrality gap, to further demonstrate the viability of integrality relaxation+quadratic regularization for MILP problems. I think some non-decision-focused "Baseline" method is still needed here to make this point.

**Audience:**

Decision-focused learning is an area of growing interest, and a new approach for differentiating through LP layers is of definite interest to the community.

**Claims And Evidence:**

Reviewers agree that the paper's theoretical results and empirical evaluations are sufficient evidence for the claims.

---

> ### Author Response · Authors · 2024-07-18
> **Response to Decision**
>
> Dear Action Editor,
>
> Thanks to you and the reviewers for this quick and highly productive reviewing process! We are glad to hear that there is excitement about the new approach, and are proud that our work was considered for additional recognition. However, due to various time constraints and other commitments of the authors, we are not currently able to do the suggested major revisions. We will implement the minor revisions and submit the camera ready version by the end of the week.
>
> We are delighted to have our work appear in TMLR!